# A multiscale modelling framework of coastal flooding events for global to local flood hazard assessments

Irene Benito[1], Jeroen C.J.H. Aerts[1,2], Philip J. Ward[1,2], Dirk Eilander[1,2], and Sanne Muis[1,2]

[1]Institute for Environmental Studies (IVM), Vrije Universiteit Amsterdam, The Netherlands.
[2]Deltares, Delft, The Netherlands.

*Correspondence to*: Irene Benito (i.benito.lazaro@vu.nl)

**Abstract.** Tropical and extratropical cyclones, which can cause coastal flooding, are among the most devastating natural hazards. Understanding coastal flood risk better can help to reduce their potential impacts. Global flood models play a key role in this process. In recent years, global models and methods for flood hazard simulation have improved, but they still present limitations to provide actionable information at local scales. One notable limitation is the insufficient resolution of global models to accurately capture the complexities of storms and topography of specific regions. Additionally, most large-scale hazard assessments tend to focus solely on either offshore water level simulations or overland flooding, often relying on static flood modelling approaches. In this study, we introduce the MOSAIC modelling framework, a flexible, Python-based framework designed to dynamically simulate both offshore water levels and coastal flooding. MOSAIC provides a multiscale modelling approach to automatically generate and nest high-resolution local models within a coarser global model. This approach seeks to simulate more accurate water levels, thereby enhancing coastal boundary conditions for dynamic flood modelling. We showcase the potential of MOSAIC for three historical storm events with the aim of assessing the effects of temporal and spatial resolution refinements and bathymetry data. Our findings indicate that the importance of model refinements is linked to the topography of the study area and the storm characteristics. For instance, refining temporal output resolution has a significant impact on small and rapidly intensifying tropical cyclones, but is less critical for extratropical cyclones. Additionally, the refinement of spatial output locations is particularly relevant in regions where water levels exhibit high spatial heterogeneity along the coast. In regions with complex topographies, grid refinement and higher-resolution bathymetry play a more significant role. MOSAIC provides an automated approach to provide flood maps at a local scale. Our results confirm the proof of concept that the automated approach of MOSAIC can be used to provide high-resolution flood maps without the need for calibration or other manual steps. As such, MOSAIC provides a bridge between fully global and fully local modelling approaches. In future work, further validation could be carried out to explore the optimal settings for different regions more in depth.

## 1    Introduction

Coastal flood events can have devastating impacts on societies, economies, and the environment when affecting densely populated and low-lying coastal areas (Wadey et al., 2015). Tropical cyclones (TCs) and extratropical cyclones (ETCs) are the cause of the most severe coastal flooding events (Douris et al., 2021; Haigh et al., 2016; UNDRR, 2020; Wahl et al., 2017). For example, Hurricane Harvey, in 2017, is one of the costliest storms in the United States' history, with an estimated damage of $125 billion. Typhoon Idai, in Mozambique 2019, caused around 600 deaths and economic damages of $770 million (Nhamo and Chikodzi, 2021; Sebastian et al., 2021). In 1953, an ETC was the cause of the most severe coastal flood event in Northwest Europe, resulting in more than 2000 deaths (Wadey et al., 2015). More recently, in 2010, ETC Xynthia hit the Atlantic coast of France, causing 47 deaths and €1.2 billion economic damages (CGEDD, 2010).

Coastal flood events are driven by extreme sea levels, resulting from a combination of mean sea level variations, tides, storm surges and waves (Kirezci et al., 2020; Marcos et al., 2019; Vousdoukas et al., 2017, 2018a; Wahl, 2017). In recent years, several studies have applied global hydrodynamic models to simulate coastal water levels (Dullaart et al., 2021; Muis et al., 2016; Pringle et al., 2021; Vousdoukas et al., 2016a; Wang and Bernier, 2023). Subsequently, these water levels have been used to derive extreme water level values for various return periods. These extreme water levels have then been used as input into global overland flood models (Wing et al., 2024), and the resulting flood hazard maps have been used to assess flood exposure and risk (Vousdoukas et al., 2016b). While these global studies have greatly improved our understanding of large-scale coastal flood risks, they do not yet have the accuracy to provide actionable information about coastal flood events at local scales.

The accuracy of large-scale hazard assessments is limited by several factors related to the quality of the input data and assumptions underlying the modelling approaches. Until now, the vast majority of large-scale hazard assessments have primarily concentrated on either modelling extreme water levels or modelling overland floods. Each model component has its own limitations. We identify here three main methodological limitations of large-scale hazard assessments. First, coastal geometry strongly influences extreme sea levels (Mori et al., 2014; Woodruff et al., 2023), with large variability at local scale. Consequently, in regions with complex morphologies, such as estuaries, semi-enclosed bays or barrier systems, global models lack the resolution required to accurately resolve the extreme sea levels (Bunya et al., 2010; Dietrich et al., 2010; Islam et al., 2021). Grid refinement and nesting of local high-resolution models within coarser global models can result in improved coastal boundary conditions. Pelupessy et al. (2017) used a multiscale approach to obtain realistic boundary conditions by nesting a global circulation model and a high-resolution barotropic model. Similarly, the Coastal Storm Modeling System (CoSMoS) combines global climate models and oceanographic models dynamically downscaled to assess compound flooding and coastal changes at regional to local scale (Barnard et al., 2025, 2019, 2014; Nederhoff et al., 2024) and Camus et al. (2011) used a dynamic downscaling approach to translate global wave data into higher spatiotemporal resolution waves for the Spanish coast. Second, the accuracy of input datasets such as the meteorological forcing and the bathymetry have large influence on the total water levels. Coarse meteorological forcings – both in terms of spatial and temporal resolution – might not be able to resolve intense storms (Hodges et al., 2017; Murakami, 2014; Thomas et al., 2021), while errors in the bathymetric datasets will propagate to the modelling of storm surge levels (Woodruff et al., 2023). Third, coastal flooding is a dynamic process where flood duration and physical processes play a key role. However, given the high computational costs associated with using hydrodynamic flood models, their use has been limited to local application. Most large-scale hazard assessments have used static flood modelling methods, which neglect flood dynamics (Hinkel et al., 2014; Muis et al., 2016; Ramirez et al., 2016; Vafeidis et al., 2019; Vousdoukas et al., 2016b). Additionally, large-scale hazard assessments typically focus on a single flood driver (Alfieri et al., 2017; Hirabayashi et al., 2021; Tiggeloven et al., 2020; Vousdoukas et al., 2018b; Ward et al., 2020). However, TC and ETC events often produce precipitation, river discharge, storm surges and waves, all of which can contribute to flooding. When these drivers occur in combinations, they can significantly amplify flood hazards and risks. This is demonstrated by the modelling of, for example, hurricane Florence that hit the US in 2018 (Gori et al., 2020). Few large-scale studies have analysed the effects and interactions of multiple flood drivers. While Bates et al. (2021) performed a combined risk assessment of fluvial, pluvial and coastal flooding for the continental USA, Eilander et al. (2023) introduced a globally-applicable compound flood modelling framework that accounts for precipitation, river discharge and storm tides.

In this study, we present the open-source MOSAIC (MOdelling Sea Level And Inundation for Cyclones) modelling framework to simulate any TC and ETC water levels and coastal flooding events. Coastal flooding is dynamically modelled by coupling of two existing modelling approaches: (1) to simulate water levels generated from storm surges and tides it couples the hydrodynamic Global Tide and Surge Model (GTSM) and Delft3D Flexible Mesh software; and (2) to dynamically simulate overland flooding it couples the simulated water levels with the Super-Fast INunadation of CoastS model (SFINCS). MOSAIC

is based on Python and global datasets, and as such provides a globally-applicable and reproducible approach that can
automatically build and process Delft3D Flexible Mesh and SFINCS models. As such it is well suited for a model comparison
study to test different model setups.
Here we showcase the potential of the MOSAIC framework by applying it to three case studies where large storm surges
caused catastrophic flooding events, namely historical storm events TC Irma, TC Haiyan, and ETC Xynthia (see Figure 1;
Bertin et al., 2012; Cangialosi et al., 2018; Lapidez et al., 2015). For each of these storms, we simulate the coastal water levels
and flood depths using automatically build, uncalibrated models. Where available, we evaluate the model performance by
comparing against observed water levels and flood maps. Moreover, we perform a sensitivity analysis of different modelling
settings. This includes the effects of model resolution, output resolution and improvements in bathymetry.

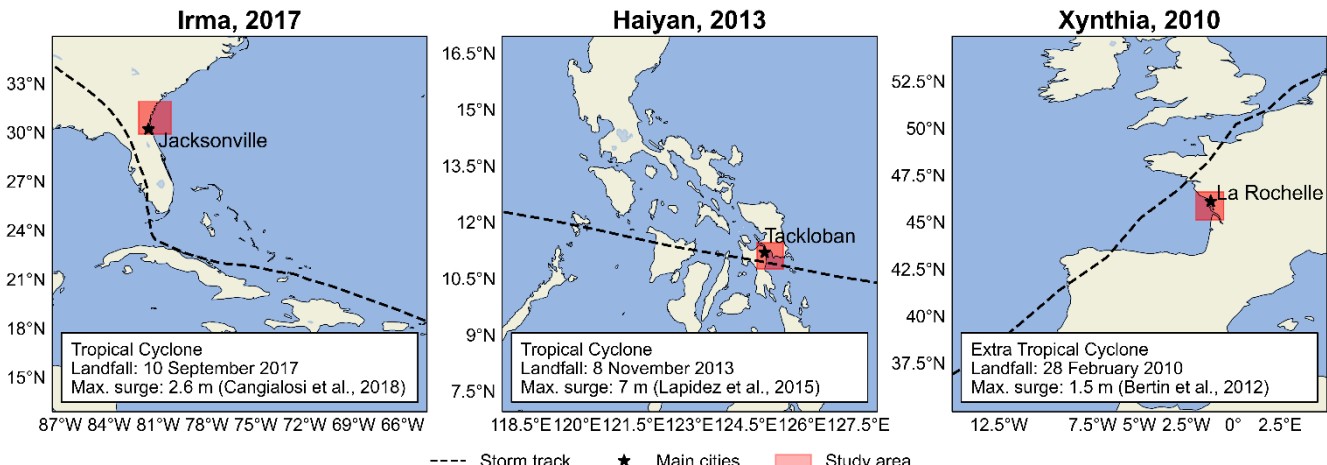


**Figure** 1**. Case studies analysed on this paper. Left: Tropical cyclone Irma; middle: Tropical cyclone Haiyan; right: Extratropical**
**cyclone Xynthia. The red area indicates the modelling domain of the flood analysis.**

## 2 The MOSAIC modelling framework

The MOSAIC modelling framework, shown in Fig. 2, is a Python-based framework that integrates different packages, models
and software. It consists of two main components: (1) the simulation of global coastal boundary conditions with the Global
Tide and Surge Model (GTSM) (Section 2.1), including the dynamic downscaling with a local high-resolution model (Section
2.1.3); and (2) the overland flood hazard simulations using the SFINCS model (Section 2.2). Python scripts that enable
adjustments to the GTSM settings are used to generate different model configurations. For the flood hazard simulations,
MOSAIC uses the Hydro Model Tools (HydroMT) to prepare and postprocess SFINCS model input and output data.

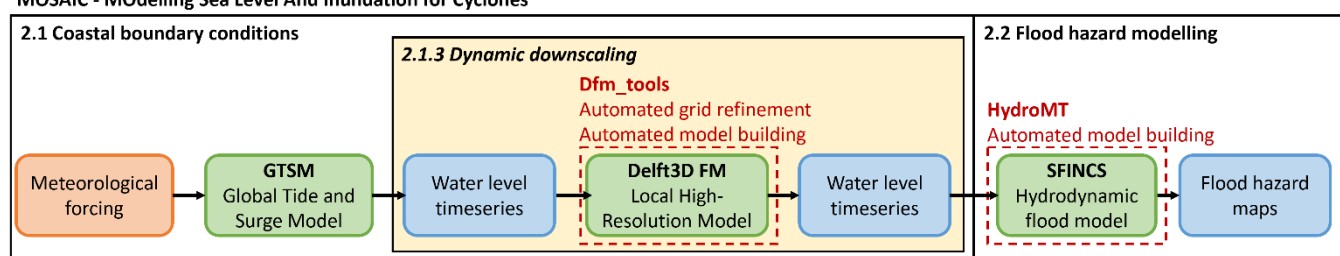

**Figure 2. Flowchart showing the input (in orange), models (in green), outputs (in blue), Python packages (in red) and the optional**
**dynamic downscaling feature (in yellow) of MOSAIC.**

## 2.1 Derivation of coastal boundary conditions

### 2.1.1 Meteorological forcing

The meteorological forcing datasets used in this study vary per storm. For ETC Xynthia and TC Irma, we use mean sea level pressure and 10 m meridional and zonal wind components from the ERA5 re-analysis dataset at a horizontal resolution of 0.25 degrees and 1 hour temporal resolution (Hersbach et al., 2019). Because TC Haiyan is not well resolved in ERA5 (see Fig. A1), we use pressure and wind from tropical cyclone track data merged with ERA5. The tropical cyclone track data is retrieved from the Joint Typhoon Warning Center at 6 hourly intervals (Naval Meteorology and Oceanography Command, 2022) and is converted to a polar grid with 36 radial bins, 375 arcs and a radius of 350 km using the Holland parametric wind model (Holland et al., 2010). Following the methodology of Dullaart et al. (2021) and Lin and Chavas (2012), we apply a counter-clockwise rotation angle of $\beta = 20°$ and set the storm translation to surface background wind reduction factor at $\alpha = 0.55$. Additionally, we use an empirical surface wind reduction factor (SWRF) of 0.85 (Batts et al., 1980), and convert 1-minute average winds to 10-minute averages using a factor of 0.915 (Harper et al., 2010). The Holland model's output provides a file that defines a polar grid containing pressure and wind fields. To extend the pressure and wind fields beyond the Holland model's defined TC boundary, we linearly interpolate these fields on the outermost 75% to align with the ERA5 background data (Deltares, 2024).

### 2.1.2 Global storm surge and tide model

MOSAIC uses GTSMv4.1 to simulate water levels resulting from tides and storm surges, ignoring baroclinic and wave contributions. GTSM is a global depth-averaged hydrodynamic model based on Delft3D Flexible Mesh (Kernkamp et al., 2011). It has a spatially-varying resolution of 25 km deep in the ocean and 2.5 km along the coasts (1.25 km for Europe) (Dullaart et al., 2020; Muis et al., 2020). The spatially-varying resolution makes it computationally efficient for simulating water levels at large scales. The bathymetry in the model is the 15 arcseconds resolution EMODnet bathymetry dataset for Europe (Consortium EMODnet Bathymetry, 2018), and the 30 arcseconds General Bathymetric Chart of Oceans 2019 dataset for the rest of the globe (GEBCO, 2014). Tides are generated internally with tide generating forces, while storm surges originate from external forcing with pressure and wind fields (Section 2.1.1; Muis et al., 2020). A constant Charnock coefficient of 0.041 is applied to translate wind speeds from the external forcing into wind drag, and a background pressure of 101,325 Pa is considered. GTSM has been successfully validated using different meteorological datasets and has been shown to provide accurate extreme sea levels (Dullaart et al., 2020; Muis et al., 2020, 2016). Version 4.1 is a calibrated version of the model with also improved parametrizations for internal tides and bottom friction coefficient (Deltares, 2021; Wang et al., 2022a). GTSM provides as output water level timeseries over a grid in the ocean and for locations along every ~5 km of the coast.

To validate the coastal component of our modelling framework, we compare water levels from GTSM against observed water levels from tide gauge stations of the Global Extreme Sea Level Analysis (GESLA) dataset (Haigh et al., 2023). This comparison is made for case studies where the GTSM output locations are found nearby tide gauge stations from GESLA (see Figure 3). GTSM output is referenced to mean sea level (MSL). We reference the GESLA water levels to the MSL by removing the annual average water level for each year, and subsequently removing the mean over the 1985-2005 period from the de-trended time series. To assess the accuracy of GTSM, we calculate the Pearson's correlation coefficient and the root mean-squared error (RMSE; see Table A1). Figure 4 and Fig. 5 show the time series of water levels at different tide gauge stations during landfall of TC Irma and ETC Xynthia, respectively. The Pearson's correlation between the GTSM-simulated and observed water levels is high for both events, indicating a good agreement. For TC Irma, the average correlation across the nine stations is 0.93 with a  standard deviation of 0.06 m. For ETC Xynthia, the average correlation across the six stations is 1.00 with a standard deviation of 0.01. Additionally, TC Irma has an average RMSE of 0.28 m with a standard deviation of 0.09 m. ETC Xynthia has a RMSE of 0.22 m with a standard deviation of 0.08 m. The stations performing less well are those located in enclosed harbours or behind the barrier islands. The RMSE values of GTSM for both storms show results comparable

to other large-scale studies that have used hydrodynamic models to simulate storm tides of storm events. Marsooli and Lin
(2018) and Gori et al. (2023), for example, used the ADvanced CIRCulation model (ADCIRC) to simulate storm tides with an
average RMSE over stations of 0.31 and 0.29 m, respectively. Vogt et al. (2024) used the GeoCLaw solver and reported an
average RMSE of 0.24 m over 213 tide gauge stations, but with a Pearson's correlation of 0.5, showing less good agreement
with observed storm tides than the MOSAIC model setup presented in this study.

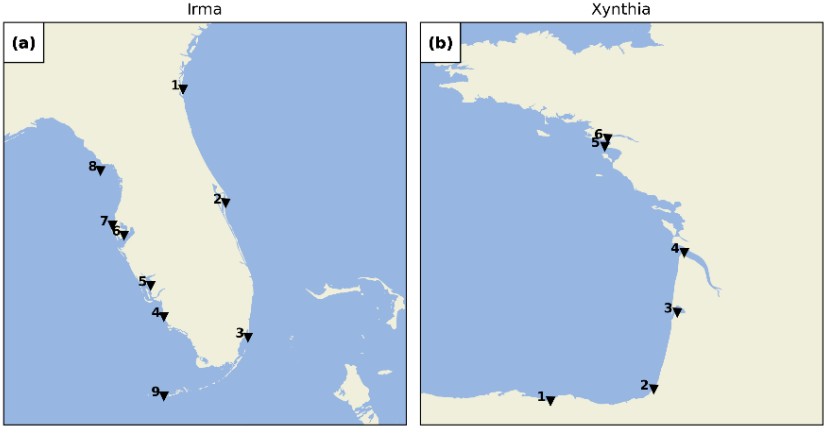


**Figure 3. GESLA tide gauge stations for the case studies Irma (panel a) and Xynthia (panel b).**

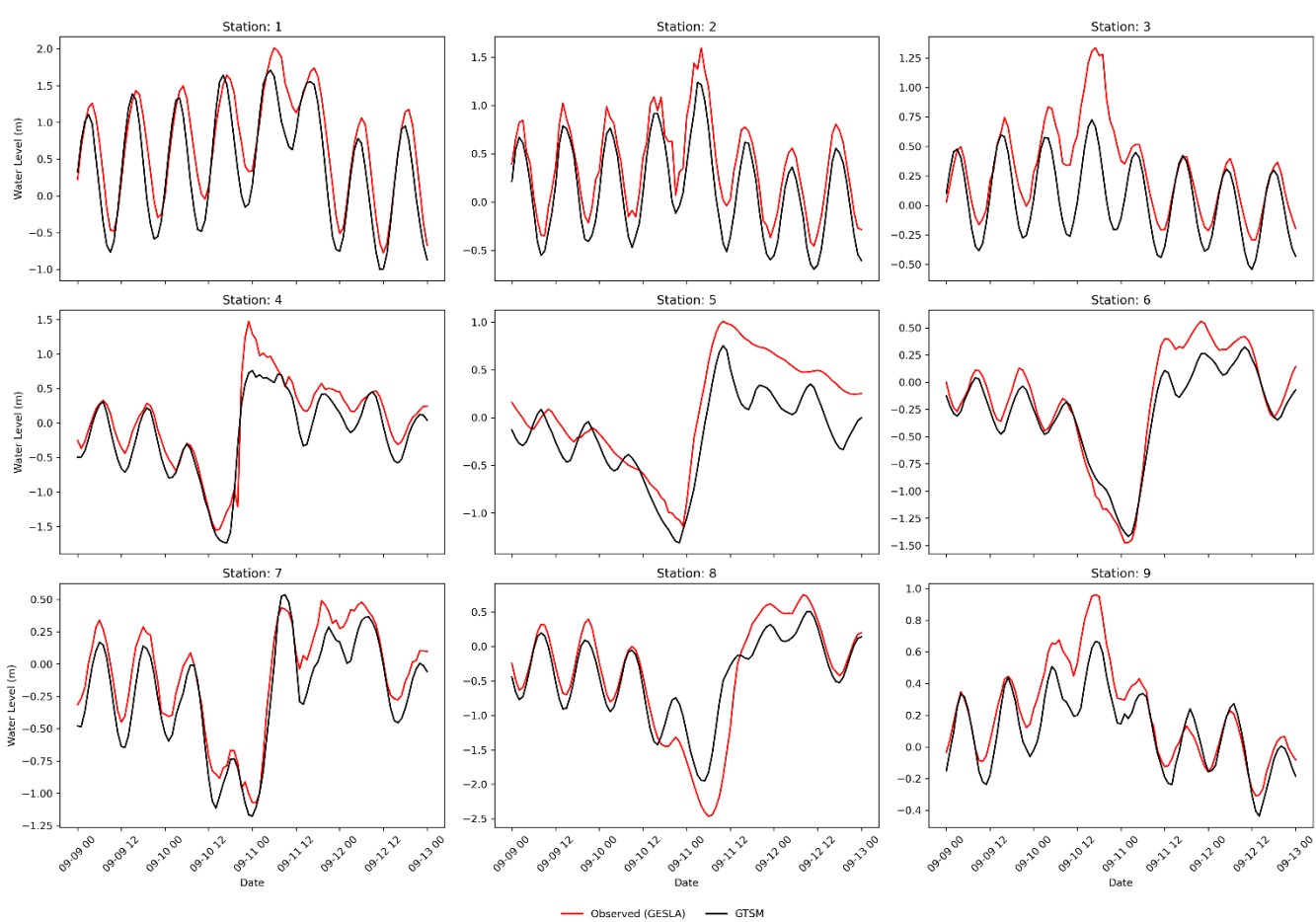


**Figure 4. Validation of water levels for the case study Irma, for the nine tide gauge stations depicted in Fig. 3.**

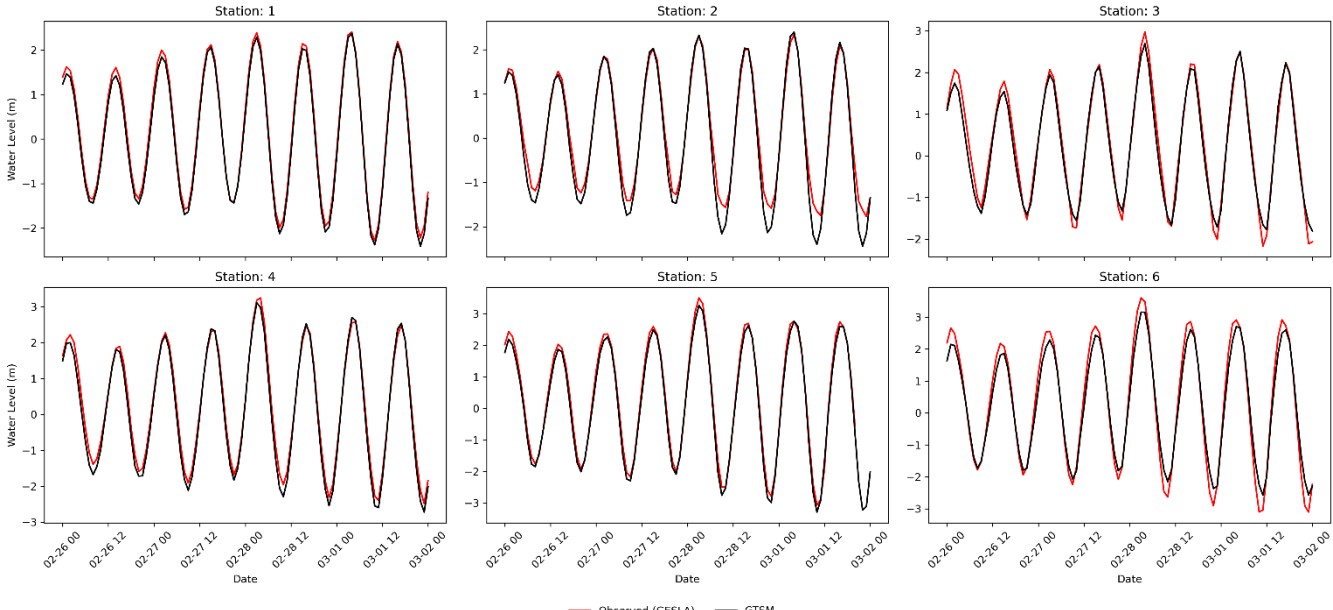


**Figure 5. Validation of water levels for the case study Xynthia for the six tide gauge stations depicted in Fig. 3.**
### 2.1.3    Dynamic downscaling
The dynamic downscaling within MOSAIC consists of two parts. First, MOSAIC generates a local high-resolution model with
Delft3D Flexible Mesh using the Python package dfm_tools (Veenstra, 2024). Dfm_tools allows to automatically create a local
modelling grid with a spatially-varying resolution based on the specified maximum and minimum grid cell sizes as well as the
Courant's number derived from the bathymetry data provided (Veenstra, 2024). The bathymetry of the local model can be
updated by interpolating a new bathymetric dataset into the newly generated grid. The settings to automatically generate the
local high-resolution models used in this study can be found in Section 2.3. Second, MOSAIC uses an offline coupling
approach to nest the local Delft3D Flexible Mesh model within GTSM. A Python script is used to first identify the boundaries
of the local Delft3D Flexible Mesh model. These boundaries are then used to determine the specific locations where GTSM
output should be extracted. Subsequently, GTSM provides the water level timeseries at the boundaries of the local model.
Finally, the local high-resolution model is executed using the water levels derived from GTSM as forcing input, together with
the same meteorological forcing as for GTSM.
## 2.2    Hydrodynamic flood hazard modelling setup
MOSAIC uses the Super-Fast INundation of CoastS (SFINCS) model to simulate overland storm surge flood depths. SFINCS
is a reduced-physics hydrodynamic model developed for a more computationally efficient dynamic flooding approach than
full shallow water equation models (Leijnse et al., 2021). It solves simplified equations of mass and momentum, similar to the
LISFLOOD-FP model (Bates et al., 2010). SFINCS has been successfully applied to model compound flooding for tropical
cyclone Irma in 2017 (Eilander et al., 2023; Leijnse et al., 2021). Its modelling output results in similar results to those from
full shallow water equation models, while reducing computational expenses by a factor of 100 (Leijnse et al., 2021). To speed
up the flood model simulations, we use the subgrid schematization from SFINCS for all the simulations (Leijnse et al., 2020).
For this study, we use GEBCO 2020 (15 arc seconds spatial resolution; (Weatherall et al., 2020)) as input dataset for the
bathymetry and FABDEM (30 m spatial resolution; (Hawker et al., 2022)) as input dataset for the land elevation. Except for
ETC Xynthia. For ETC Xynthia we use the 5 m resolution LiDAR-based DEM developed by the French National Geographic
Institute (IGN) because it better represents dikes in the region, leading to better flood estimates than FABDEM (see Fig. A14).
The spatially varying roughness coefficients used within SFINCS are derived from the land use maps of the Copernicus Global
Land Service (Buchhorn et al., 2020). Within MOSAIC, SFINCS is coupled offline with water levels from GTSM at 1-hourly

resolution for the default settings. The Mean Dynamic Topography (DTU10MDT; (Andersen and Knudsen, 2009) is used to convert the vertical reference of the water levels from mean sea level to the EGM2008 geoid. The resulting flood hazard maps have a resolution of 30 m.

To build the SFINCS models and couple them with GTSM, MOSAIC uses the HydroMTv0.7.1 (Hydro Model Tools) package (Eilander et al., 2023). HydroMT is an open-source Python package, which provides automated and reproducible model building and analysis of results. HydroMT uses a modular approach in which datasets and model setup configurations can easily be interchanged. In the MOSAIC framework presented in this paper, we take advantage of HydroMT in several ways: (1) to automatically convert the forcing files from GTSM and the other input into the model specific input format; (2) to easily build a reproducible SFINCS model; and (3) to perform the analysis of the SFINCS model output. SFINCS is forced with GTSM water level timeseries at locations along every ~5 km of the coastline, and provides as output water level timeseries for each grid cell. Finally, flood depth maps are derived from the maximum water levels by subtracting the DEM.

To validate the hydrodynamic flood hazard modelling component of the modelling framework, we compare the modelled flood extents with observed flood extents derived from field measurements. This comparison is done for Xynthia, the only case study for which observed flood extent data are available (Breilh et al., 2013; DDTM, 2011). We measure the model skill using: (1) the hit rate (H), defined as the flood area correctly simulated over the observed flooded area (Eq (1)); (2) the false-alarm ratio (F), defined as the area wrongly simulated over the observed flooded area (Eq (2)); and (3) the critical success index (C), defined as the area correctly simulated to be flooded over the union of the observed and modelled flooded area (Eq (3)). Figure 6 shows the skill of the modelled maximum flood extents by SFINCS using the GTSM water levels as forcing. The hit rate is 0.78, correctly representing the flooding in most regions, only underestimating it in regions further inland. The false-alarm ratio of the model is 0.62. Flooding is overestimated in the north, likely due to the lack of flood protection measures included in the model that are present in reality. The critical success index is 0.48, as a result of the areas well simulated and those over and underpredicted. While the performance of the flood model is negatively affected by the quality of the topography and the representation of local features such as dikes, we consider the performance sufficient for large-scale modelling and comparable to other studies such as Ramirez et al. (2016) and Vousdoukas et al. (2016b).

$$H = \frac{F_{modelled} \cap F_{observed}}{F_{observed}} \tag{1}$$

$$F = \frac{F_{modelled} / F_{observed}}{F_{observed}} \tag{2}$$

$$C = \frac{F_{modelled} \cap F_{observed}}{F_{modelled} \cup F_{observed}} \tag{3}$$

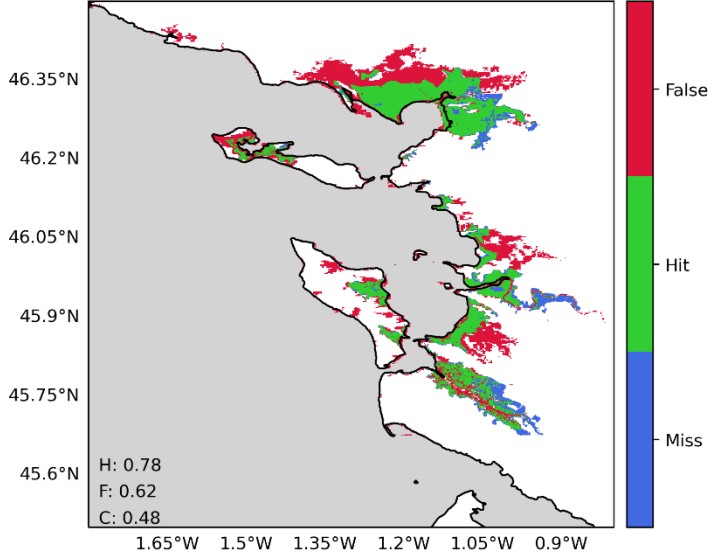

**Figure 6. Validation of the flood hazard modelling component of the modelling framework for the case study Xynthia, using the water levels of the default configuration of GTSM as a forcing. The maps compare the modelled and observed maximum flood extents, where: green indicates flood areas correctly simulated; blue flood areas not simulated but observed; and red flood areas simulated but not observed. Performance indicators for the hit rate (H), false-alarm ratio (F) and critical success index (C) are shown in the panel.**

## 2.3    Sensitivity analysis

Using the MOSAIC modelling framework, we analyse the effects of refining the resolution of GTSM on the simulated water levels and assess how these propagate into the results for the flood hazard simulated by SFINCS. As described in Table 1, we categorise model configurations in two distinct groups. The first group, which contains the global model configurations (G), includes the default model configuration (G1) and configurations that modify only the global GTSM model (G2 and G3). In this group, the refinements applied are: (1) the temporal output resolution, which is different than the implicitly calculated simulation timestep of GTSM, is refined from 1-hourly to 10-minute, allowing to capture more changes in water levels, including the peaks of the water levels (G2); and (2) the spatial output resolution is refined from locations along the coast every ~5 km to ~2 km, providing more coastal boundary conditions for the hydrodynamic flood hazard model (G3). The second group, which contains the nested model configurations (N), includes those model configurations that use a nested local model within the global model GTSM by performing dynamic downscaling. These model configurations include: (1) the nesting of local high-resolution models with refined grids into GTSM (N1); and (2) the nesting of local high-resolution models with refined grids and updated bathymetry into GTSM (N2). Finally, we evaluate the combined effects of all these refinements through the "fully refined" configuration (N3), which integrates both the enhanced temporal and spatial resolutions as well as the nested high-resolution models and updated bathymetry. The validation of GTSM and SFINCS shows sufficient performance for all the model configurations from Table 1 and Fig. 7 (see Table A1 and Figs. A2, A3 and A15).

**Table 1. GTSM model configurations used in the sensitivity analysis.**

| Model configuration | Nomenclature | GTSM grid resolution | Bathymetry | Spatial output resolution | Temporal output resolution |
|---|---|---|---|---|---|
| **Default configuration** | G1 | ~25 to 2.5/1.25km | GEBCO2019 * | Original (~5 km) | 1h |
| **Refined temporal output resolution** | G2 | ~25 to 2.5/1.25km | GEBCO2019 * | Original (~5 km) | 10min |
| **Refined spatial output** | G3 | ~25 to 2.5/1.25km | GEBCO2019 * | Refined (~2 km) | 1h |
| **Dynamic downscaling (Refined grid)** | N1 | ~25 to 0.45km | GEBCO2019 * | Original (~5 km) | 1h** |
| **Dynamic downscaling (Refined grid + Updated bathymetry)** | N2 | ~25 to 0.45km | GEBCO2023 | Original (~5 km) | 1h** |
| **Fully refined configuration** | N3 | ~25 to 0.45km | GEBCO2023 | Refined (~2 km) | 10min** |

* EMODnet2018 for Europe (Xynthia case study)
**For the model configurations N1, N2 and N3, the temporal output resolution is also the temporal resolution of the coupling between
GTSM and the local high-resolution model.

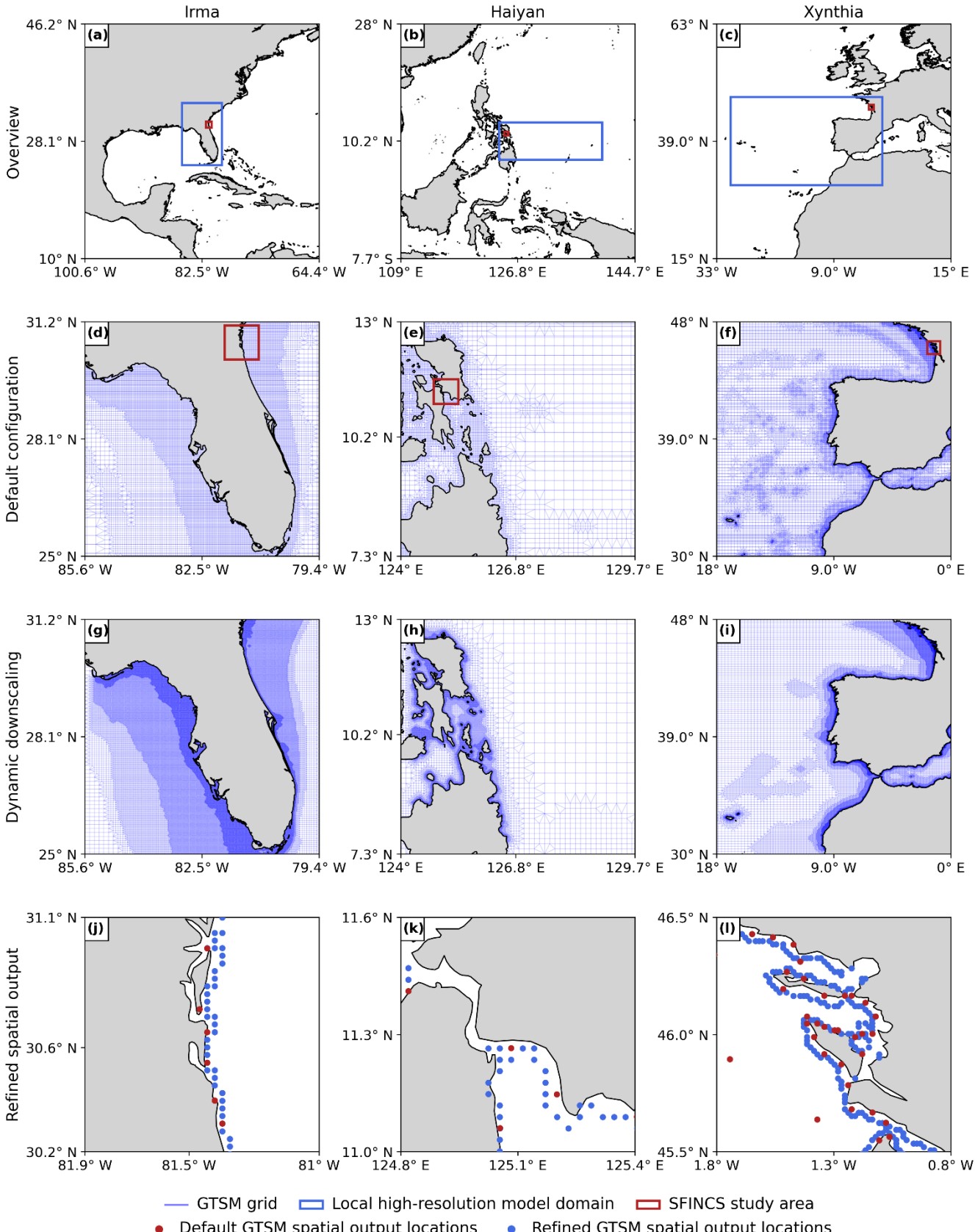


**Figure 7. Overview of the model domains for the local high-resolution model and SFINCS, for the three case studies (panels a, b, c);**
**default GTSM grid zoomed in (d, e, f); local high-resolution model grid zoomed in (g, h, i) and; GTSM spatial output locations for**
**the default configuration and the refined spatial output configuration, zoomed into the SFINCS study area (j, k, l).**

### 3  Sensitivity analysis of the model results

#### 3.1  Multiscale storm surge modelling

Figure 8 panels a, e and i show the maximum water levels simulated by G1 for the three case studies, and depict the maximum observed water levels for various GESLA tide gauge stations. To understand the effect of each individual refinement in the maximum water levels, Figure 8 presents the differences in maximum water levels between each refinement and the model configuration G1. Figure 9 presents the differences in maximum water levels between the fully refined model configuration N3 and the model configuration G1.

#### 3.1.1  Effects of higher resolution on water levels

Figure 8 panels b, f, j show that the refinement of temporal output resolution of GTSM from 1-hourly to the 10-minute intervals of G2 results in higher maximum water levels across the entire model domain for all three case studies. For TC Irma (Fig. 8 panel b), the sensitivity of the water levels to the temporal refinement is relatively small, less than 10 cm. The small effect of the temporal refinement for TC Irma can be observed as well in Table A1 and Fig. A2, where G1 and G2 present similar timeseries and performance coefficients when compared to observed water levels. For TC Haiyan (Fig. 8 panel f), the sensitivity of the water levels is significant. Water levels increase due to the temporal refinement up to 2 m along the coastlines where TC Haiyan made landfall, showing that 1-hourly resolution is too coarse to accurately capture the water level response. The cause for this is that TC Haiyan had a rapid intensification, and when modelling water levels at 1-hourly resolution we overlook the storm's peak, resulting in an underestimation of the maximum water levels. G2 however, can capture the peak of TC Haiyan more precisely (see Figs. A4 and A5). For ETC Xynthia (Fig. 8 panel j), the sensitivity of the water levels to the temporal refinement is relatively small, less than 10 cm on average, and slightly higher in enclosed basins and estuaries near La Rochelle. The small changes in water levels for ETC Xynthia are due to the inherent characteristics of ETCs, which typically have larger dimensions, lower intensity, and a slower rate of intensification compared to TCs. This means that the changes in water levels can be well captured at a 1-hourly resolution. The small effect of the temporal refinement for ETC Xynthia can be observed as well in Table A1 and Fig. A3, where G1 and G2 present similar timeseries and performance coefficients when compared to observed water levels.

The model configuration G3, where the spatial output resolution is refined, is not shown in Fig. 8 because increasing the number of water level locations does not change the water level values themselves. However, this refinement becomes significant when these values are applied as coastal boundary conditions to SFINCS (see Section 3.2.1), as a greater number of coastal boundary conditions offer additional information for the flood model.

#### 3.1.2  Effects of dynamic downscaling with original bathymetry on water levels

Figure 8 panels c, g, k show that the model configuration N1 results in significant changes in water levels for all case studies. The largest differences occur along the coasts, where the largest changes in model grid size resolution occur. For TC Irma (Fig. 8 panel c), the nesting of a local model at high-resolution with GEBCO2019 results in maximum water levels that are up to 0.3 m higher than G1 in the southwest of Florida. These differences between N1 and G1 gradually increase over time and are maximum at the peak of TC Irma (Fig. A10). While higher grid resolution affects the tidal propagation mainly along the coast of Florida (Fig. A6 and Figure A7), storm surge propagation is more sensitive to the used bathymetry (Fig. A8 and Figure A9). High resolution is needed in areas with steep bathymetry. In contrast to the coarser grid of G1, N1 better resolves complex topographic features around the barrier islands (Fig. A11), allowing water to flow more freely through these barriers. At timestep 10-09-2017 in Figure A10, when there is a negative surge north of the barrier island, G1 produces higher water levels because water remains trapped in the north. Conversely, during the peak of TC Irma, on the 11-09-2017, the water levels in G1 are lower than N1 because less water is able to travel northwards. The increased northward surge of N1 propagates further into the Gulf of Mexico, leading to higher water levels that also propagate further into the Gulf of Mexico (see Figure A10). Water levels for nine tide gauge stations along the coast indicate that while G1 underestimates the peak of TC Irma in most

locations (Fig. A2, all stations but station 7), N1 simulates on average higher peaks, resulting sometimes in overestimations (Fig. A2, station 9). The improved resolution of topographic features in the barrier island region allows stations nearby (Fig. A2, stations 4 and 9) to better capture the event's peak compared to G1. Additionally, the performance of N1 is slightly better than G1 for six tide gauge stations (stations 1-6), as reflected in Table A1, which shows lower RMSE values. However, for stations 7-9, G1 shows slightly higher RMSE and Pearson's correlation. For TC Haiyan (Fig. 8 panel g), the differences in maximum water levels are up to 1 m higher than G1 near the landfall regions. These differences occur due to the refinement of the grid from 2.5 km to 45 m, which results in a significant increase in the number of model grid cells that define regions of shallow bathymetry, especially around the bay near Tacloban, resulting in a more detailed representation of water levels in that region. Thanks to the increase on grid cells, the strait north of Tacloban for N1 is defined with multiple grid cells in comparison to the two grid cell width of G1 (see Fig. A12). Therefore, in that region N1 allows us to better resolve the topography of the region, and water can travel more easily northwards. For ETC Xynthia (Fig. 8 panel k), the water levels from the nested local model at high-resolution are overall lower than water levels for the G1. Near La Rochelle, those water levels are up to 0.2 m lower. When comparing the performance of N1 with G1 (Table A1 and Fig. A3), both model configurations can predict the timeseries pattern well, with high Pearson's correlation coefficients. Overall, the RMSE for Xynthia is similar for most tide gauge stations, except for two stations located in the mouth of estuaries (stations 3 and 6).

### 3.1.3 Effects of dynamic downscaling with updated bathymetry on water levels

Figure 8 panels d, h, l show that the model configuration N2 results in relatively large changes in the water levels for all the case studies. The largest differences occur along the coasts and provide figures similar to those from N1. For TC Irma (Fig. 8 panel d), the nesting of a local model at high-resolution with updated GEBCO2023 bathymetry results in maximum water levels that are 0.3 m higher than G1 in the south of Florida. Compared to N1, model configuration N2 provides slightly higher water levels south of Florida. Those differences come from differences between GEBCO2023 and GEBCO2019 in the region. N2 shows a similar performance to G1 and N1 across nine tide gauge stations (Table A1 and Fig. A2). For TC Haiyan (Fig. 8 panels h), the differences in maximum water levels are up to 1 m higher than G1 at the landfall regions. Compared to N1, N2 provides on average higher maximum water levels, except in the bay of Tacloban where N1 presents on average higher maximum water levels. These differences come from the differences in GEBCO2019 and GEBCO2023. For ETC Xynthia (Fig. 8 panels l), the water levels from the nested local model at high-resolution with GEBCO2023 are lower overall than water levels for G1. Compared to N1, the model configuration N2 provides a similar pattern of water level decrease, however, the maximum water level reduction compared to G1 is slightly less than for N1. The performance of N2, as shown in Table A1 and Fig. A3, is comparable to that of G1 and N2, except at two tide gauge stations (station 3 and 6) where GEBCO2023 does not accurately capture the bathymetry of the river channels in the estuaries. In contrast, EMODNET2018, the bathymetry used in model configuration N1, better resolves these details (see Fig. A13).

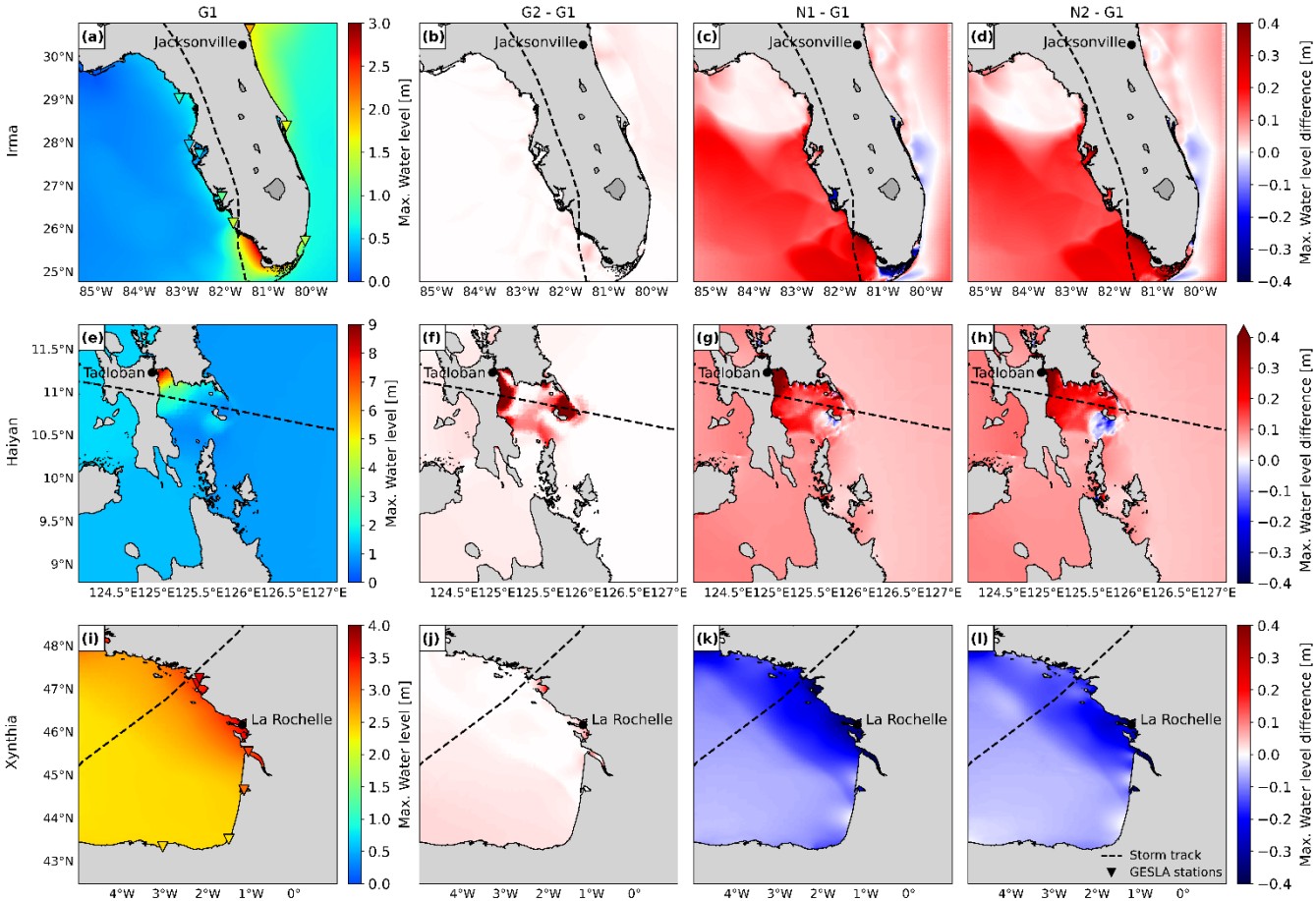


**Figure 8. Maximum water levels for the three case studies for G1 (panels a, e, i). Difference between the maximum water level for**
**each specific model configuration (see Table 1) and G1. Panels a, e, i show observed maximum water level from tide gauge stations**
**of GESLA. Difference in water levels for G2 (panels b, f, j), N1 (panels c, g, k) and N2 (panels d, h, l).**

### 3.1.4 Effects of a fully refined model on water levels

In Fig. 9 we observe that the maximum water level differences between N3 and G1 lead to significantly different results for
each case study. For TC Irma N3 provides higher maximum water levels throughout almost the whole the domain, resulting
in a picture similar to N2 but with higher water levels along the southeast coast. The maximum differences in maximum water
levels between N3 and N1 are up to 0.3 m. For TC Haiyan N3 provides maximum water levels that resemble a combination of
G2 in the regions where temporal refinement is relevant, and N2 in the rest of the study area. The differences between N3 and
G1 in maximum water levels for Haiyan are more than 2 m in the coast near Tacloban. Finally, for ETC Xynthia N3 provides
slightly higher maximum water levels in the south of the domain compared to G1, where the effects of G2 predominate, and
lower maximum water levels in the north, where the effects of N2 are more dominant.

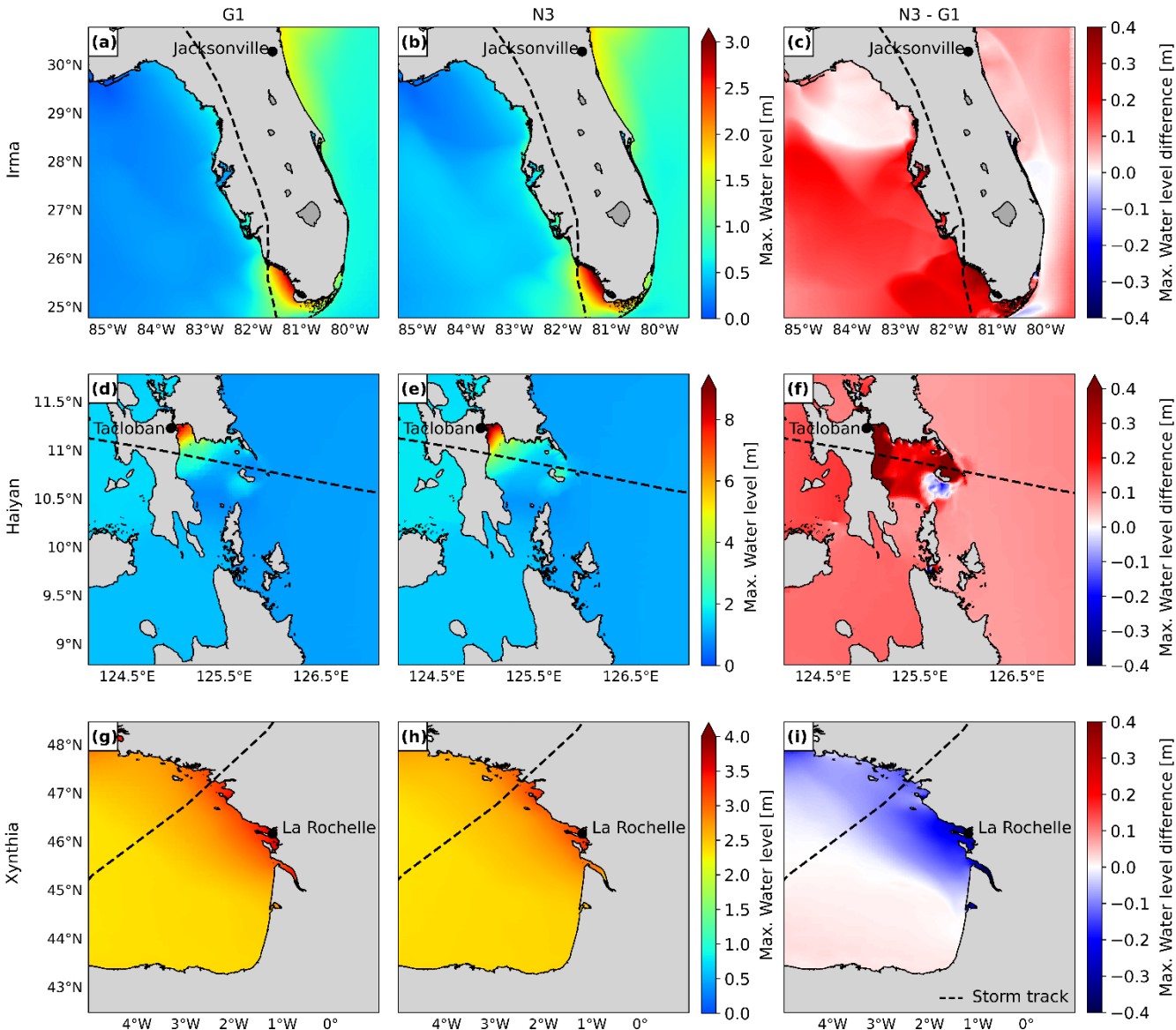

**Figure 9. Maximum water levels for the three case studies, for the default configuration G1 (panels a, d, g) and for the fully refined configuration N3 (panels b, e, h). Difference between the maximum water level for N3 model configuration and G1 (panels c, f, i).**

### 3.2 Hydrodynamic flood modelling

As a second step in the sensitivity analysis, we analyse how the effects of the different storm surge model configurations propagate to the SFINCS flood model. In Figure 10 we compare the maximum flood depths of each refinement and G1. Figure 11 shows the maximum flood depth differences between N3 and G1.

#### 3.2.1 Effects of higher resolution on flood depths

Figure 10 panels b, g, l show that the refinement of GTSM's temporal output resolution from 1-hourly to 10-minute intervals of G2 provides different results for each case study. For TC Irma (Fig. 10 panel b), the small increase in water levels as a result of the temporal output refinement (Section 3.1.1) also results in a small increase in flood depths. Conversely, TC Haiyan (Fig. 10 panel g) experiences much higher water levels along the coast at higher temporal resolution. As a result, it also experiences significantly higher flood depths, surpassing G1 by 1m in regions near Tacloban. ETC Xynthia (Fig. 10 panel l) experiences an increase in water levels along the coast for the 10-minute temporal output resolution, especially in the study region of SFINCS. This results in an increase in flood depths of up to 0.1 m. For ETC Xynthia, G2 shows a higher hit rate and false-alarm ratio compared to G1, but the same critical success index (see Fig. A15).

Figure 10 panels c, h, m show that refinement of the spatial output locations of G3 provides coastal boundary conditions to SFINCS at additional locations, thereby providing more water level input to the flood model. Figure 10 panel c shows that this refinement results in lower flood depths north and around Jacksonville for TC Irma. Conversely, for TC Haiyan (Fig. 10 panel h), the increase in spatial inputs results in higher flood depths in most of the study area, particularly exceeding more than 1 m the G1 flood depths around Tacloban. For ETC Xynthia (Fig. 10 panel m) the refinement of spatial water level inputs leads to higher flood depths north of La Rochelle of up to 0.1 m, while south of La Rochelle there are barely any changes compared to G1. For ETC Xynthia, G3 shows the same hit rate as G1, higher false-alarm ratio and the same critical success index (see Fig. A15).

### 3.2.2 Effects of dynamic downscaling with original bathymetry on flood depths

Figure 10 panels d, i , n show that the model configuration N1 results in significant changes in the flood depths for all the case studies. For TC Irma (Fig. 10 panel d), model configuration N1 leads to slightly higher water levels in comparison to G1. Consequently, the resulting flood depths are also larger and are more than 0.2 m above those of G1. Maximum water levels for TC Haiyan (Fig. 10  panel i) are generally higher along the bay of Tacloban when applying dynamic downscaling with the original bathymetry. This results on average in higher flood depths of more than 1 m compared to G1. Finally, ETC Xynthia (Fig. 10 panel n) presents lower water levels for N1 compared to G1. Those lower water levels lead to lower flood depths across the whole model domain. For ETC Xynthia, N1 shows a lower hit rate and false-alarm ratio compared to G1, and the same critical success index (see Fig. A15).

### 3.2.3 Effects of dynamic downscaling with updated bathymetry on flood depths

Figure 10 panels e, j, o show that the model configuration N2 results in significant changes in flood depths for all case studies. For TC Irma (Fig. 10 panel e), model configuration N2 compared to G1 leads to higher and lower water levels, depending on the region. Consequently, the resulting flood depths for N2 vary between 0.05 m lower to more than 0.2 m higher than G1. Maximum water levels for TC Haiyan (Fig. 10 panel j) are generally higher in the bay of Tacloban for model configuration N2 (when applying dynamic downscaling with the updated bathymetry) compared to G1. This results in larger flood depths which, in some regions, result in more than 1 m higher compared to G1. However, in the Tacloban Bay N1 results on average in higher maximum water levels than N2, which leads to lower flood depths for N2 in comparison to N1. Finally, for ETC Xynthia (Fig. 10 panel o) water levels are lower for N2 compared to G1. Those lower water levels lead to lower flood depths across the whole model domain. For ETC Xynthia, N2 shows a lower hit rate and false-alarm ratio compared to G1, and the same critical success index (see Fig. A15).

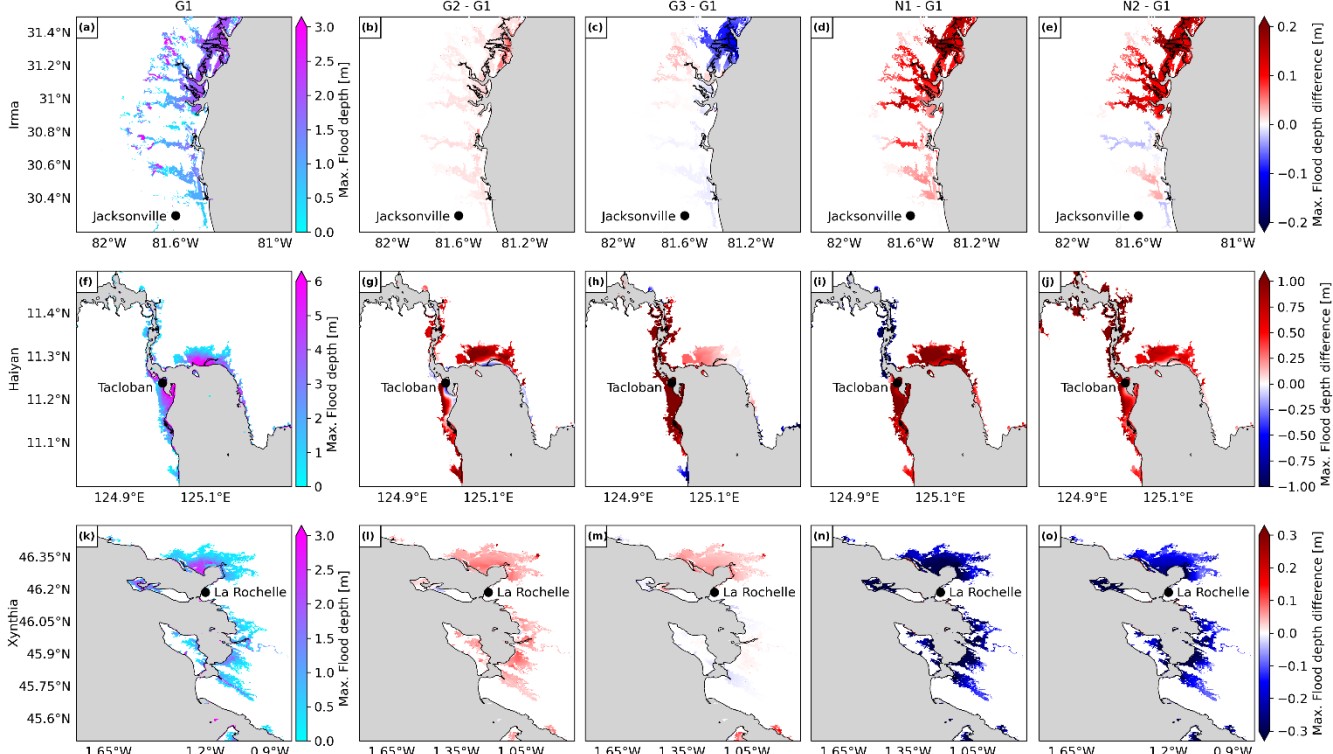

372

**Figure 10. Panels a, f, k show the maximum flood depth for the default configuration G1, for each case study. Panels b, g, l show the difference between the maximum flood depth for the refined temporal output resolution configuration G2 and G1. Panels c, h, m show the difference between the maximum flood depth for the refined spatial output configuration G3 and G1. Panels d, i, n show the difference between the maximum flood depth for the dynamic downscaling (refined grid) configuration N1 and G1. Panels e, j, o show the difference between the maximum flood depth for the dynamic downscaling (refined grid and updated bathymetry) configuration N2 and G1.**

### 3.2.4    Effects of a fully refined model on flood depths

For TC Irma N3 provides higher water levels throughout large parts of the domain (Section 3.1.4) that translate into higher flood depths up to more than 0.2 m near Jacksonville. For TC Haiyan, N3 provides high water levels near Tacloban (Section 3.1.4), translating into high flood depths up to more than 1 m. Finally, ETC Xynthia presents lower water levels for N3 near La Rochelle (Section 3.1.4), which translate into lower flood depths along the coast.

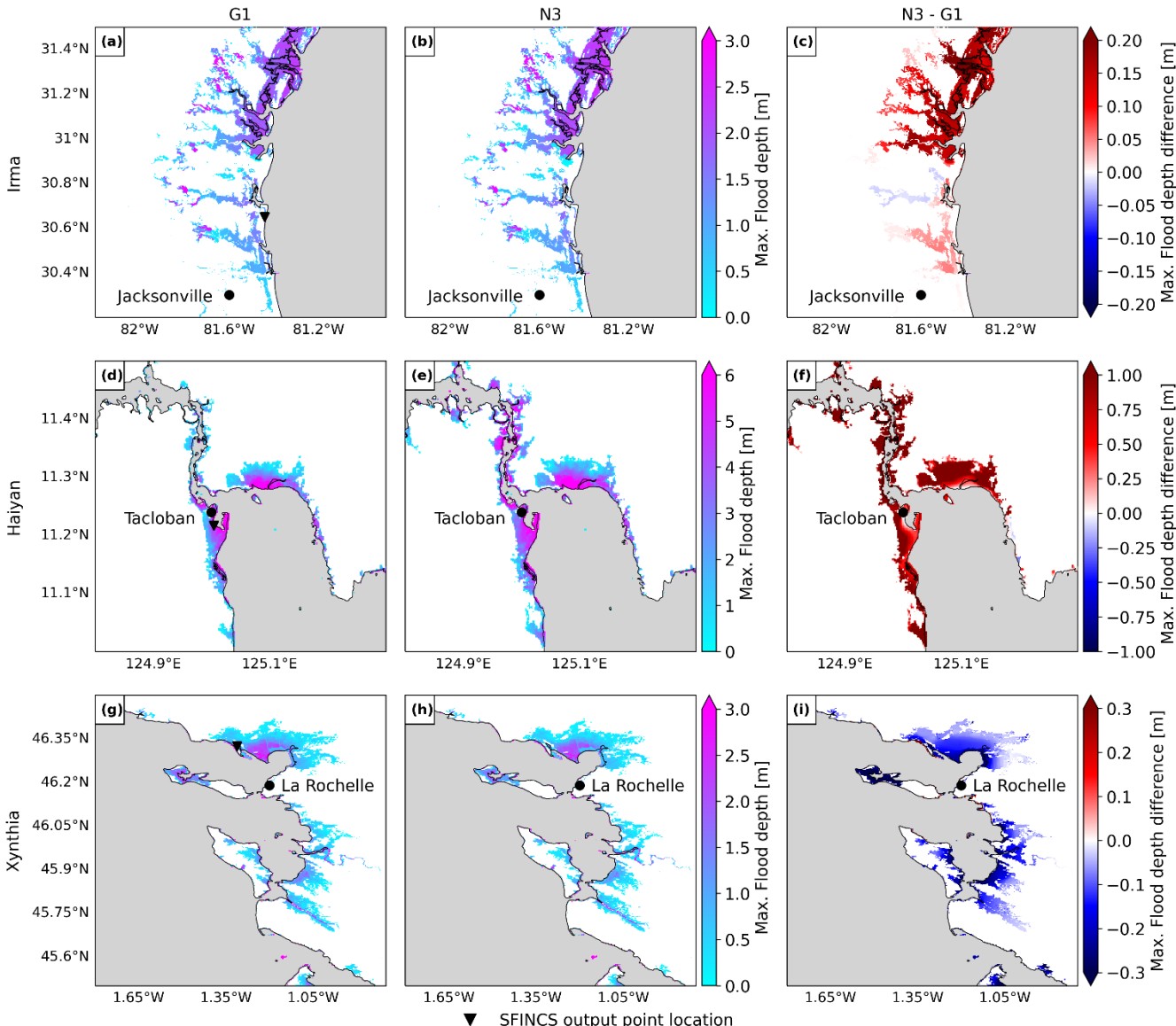

385

**Figure 11. Panels a, d, g show the maximum flood depth for the default configuration G1 for each case study. Panels b, e, h show the maximum flood depth for the fully refined configuration N3. Panels c, f, i show the difference between the maximum flood depth for N3 and G1.**

To analyse the changes of flood depths over time, Fig. 12 panels a, b, c show the flood depth timeseries at the SFINCS output point locations outlined in Fig. 11, for all the model configurations. The timing and shape of the flood depth timeseries remain consistent across all the model configurations for all the case studies, an only slight differences in the magnitude of the flood depths are visible. Figure 12 panel a shows that for TC Irma all the model configurations result in similar flood depths, and only N1 results in slightly higher flood depths of about 0.1 m more than the others. Figure 12 panel b shows that for TC Haiyan G1 results in the lowest flood peak, while the temporal resolution of G2 plays a key role, enhancing the flood peak up approximately 1 m higher than G1. Finally, Fig. 12 panel c shows that for all global model configurations (G1, G2 and G3) result in a first flood peak that is approximately 0.5 m higher than those of the nested model configurations (N1, N2 and N3). The second peak is simulated more similarly by all model configurations, being N1 the configuration that provides lowest flood depths.

Panels a, b, c in Fig. 12 only show the results for a single SFINCS output point location. However, the refinements might have most effect in other regions different than the SFINCS output point locations. To understand the overall effect of each

refinement in the flood hazard maps, Fig. 12 panels d, e, f show the flood volume timeseries across each of the case study's model domain. While the timing and shape of the flood volume timeseries remains consistent across all the model configurations for all the case studies, there are differences in the magnitude of the flood volumes. Figure 12 panel d shows that for TC Irma the nested models lead to the highest flood volumes, being N3 the model configuration that simulates the highest flood volume. On the other hand, the increase in spatial output of GTSM from G3 results in the lowest flood volumes. Figure 12 panel e shows that for TC Haiyan N3 also leads to the highest flood volumes, while G1 results in the lowest volumes. Finally, Fig. 12 panel f shows that for ETC Xynthia the nested model configurations lead to the lowest flood volumes, while the global models result in higher flood volumes.

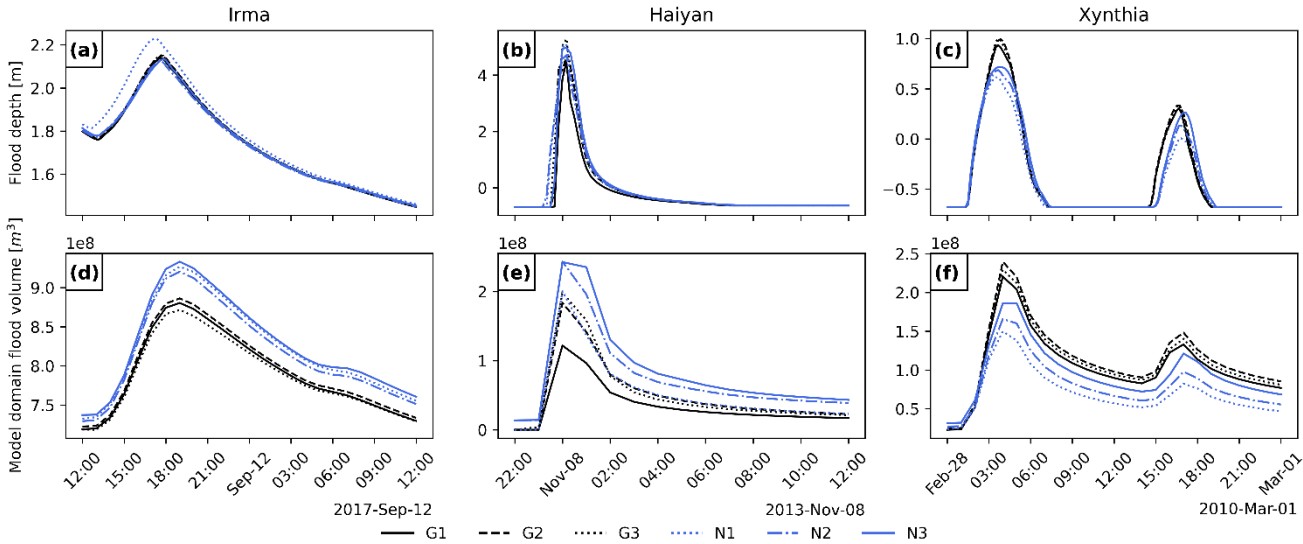

**Figure 12. Flood depth timeseries for three observation points and flood volume timeseries for the SFINCS model domain of each case study and model configuration (see Table 1). The spatial location of the SFINCS output point locations can be observed in Fig. 11 panels a, d, g.**

## 4    Discussion

### 4.1    Sensitivity analysis and model validation

The results of the sensitivity analysis reveal the complexity of hydrodynamic modelling and the sensitivity to specific local settings and storm characteristics. The effect of nesting higher resolution models on water level and flood depth varies. For instance, the fully refined model configuration N3 simulates higher water levels almost everywhere for TC Irma. However, for TC Haiyan and ETC Xynthia, certain regions show higher water levels with N3, while other regions show lower water levels compared to the default global G1 configuration. Similarly, flood depths around Jacksonville for TC Irma are generally higher with the refined model configuration N3, although some areas experience lower values. For TC Haiyan in Tacloban, flooding significantly increases with the refinements, whereas for ETC Xynthia flood depths decrease notably around La Rochelle.

Refining the temporal output resolution (model configuration G2) has a significant influence on small, rapidly intensifying TCs, like Haiyan. Compared to the default global configuration G1, this results in water levels and flood depths that are 2 m and 1 m higher. For ETCs, the refinement of temporal output resolution does not lead to substantial changes in water levels or flood depths, indicating that a 1-hourly temporal resolution is sufficient. Refining the spatial output locations of GTSM (model configuration G3) provides more detailed coastal boundary conditions for SFINCS. This is most relevant for regions where the coastal water levels show large spatial variations. For TC Haiyan, for example, the increase of coastal output locations in the bay of Tacloban from 4 to 20 location (see Fig. 7), leads to flood depths 1 m higher than G1. Furthermore, regions with more complex topographies such as the south of Florida for TC Irma or the Tacloban bay for TC Haiyan are influenced by the

grid refinement of N1, leading to larger differences with G1 in terms or water levels and consequently, flooding. The updating of bathymetry also plays an important role, contributing to differences between N1 and N2 in all the case studies.

The validation of our results also highlights the complexities of hydrodynamic modelling, and how each specific setting impacts overall performance. It is challenging to assess the storm surge model performance due to the limited number of tide gauge stations available with poor spatial coverage in many regions (Haigh et al., 2023). Another source of uncertainty is the location of these tide gauge stations, which are often situated in enclosed basins or harbours, where hydrodynamic models have more difficulty simulating water levels compared to open sea conditions. Besides, the validation of the flood hazard models is difficult due to the contribution of other flood drivers neglected in this study. The automated, uncalibrated MOSAIC configurations tested in this study have a storm surge modelling performance with Pearson's correlations above 0.92 and average RMSE less than 0.3 m. These results are comparable to the well-established GTSM model (Muis et al., 2016) and to other large-scale studies (Gori et al., 2023; Marsooli and Lin, 2018; Vogt et al., 2024). Similarly, the flood hazard modelling results align with those from other studies that simulated coastal flooding from ETC Xynthia (Ramirez et al., 2016; Vousdoukas et al., 2016b). All model configuration refinements perform adequately, with similar results, making it difficult to determine which configuration consistently provides the best overall performance based on the validation. This outcome largely depends on the storm characteristics and regional topography.

### 4.2 Limitations

There are several limitations that need to be taken into account when using MOSAIC. Limitations that are linked to general flood hazard modelling and not specific to MOSAIC include the following: (1) the meteorological forcing data can be a large source of uncertainty when modelling extreme water levels (Dullaart et al., 2020). MOSAIC allows to combine the results of the Holland parametric wind model with climate reanalysis datasets to enhance the wind and pressure fields at the peripheries of the TCs. Nonetheless, the implementation of more advanced parametric wind models or high-resolution climate data could further improve the water level simulations (Emanuel and Rotunno, 2011; Hu et al., 2011). (2) the accuracy of the bathymetry has a large influence on storm surge modelling (Mori et al., 2014; Woodruff et al., 2023). Global bathymetry is rather coarse and can have large errors (Weatherall et al., 2020), but for many regions high-resolution and accurate bathymetry is not available. This will impact the effect of dynamic downscaling, where MOSAIC uses bathymetry data to generate the model grid. Using higher-resolution local bathymetry enables finer grid refinement and higher accuracy of local data (Consortium EMODnet Bathymetry, 2018; NOAA, 2014; NOAA National Geophysical Data Center, 2001), which can enhance the accuracy of the results (Woodruff et al., 2023). (3) the accuracy of digital elevation models (DEMs) has a large influence on flood modelling (Hawker et al., 2022). In this paper we use FABDEM and IGN, but MOSAIC allows to replace the DEM with any dataset, and we recommend users of MOSAIC to use the best data available for their region of interest. In addition to the effects of DEMs, the presence of flood protection structures has substantial impact on flood hazard models. The neglection of dikes in our SFINCS model is one of the reasons our modelling framework overestimates flooding for ETC Xynthia. MOSAIC's HydroMT component supports the implementation of levees as 1D line features into the SFINCS model, and this capability could be used when there is local information on flood protection levels.

The main limitation specific to the automated approach of MOSAIC is related to the generation of the local high-resolution models. These automatically generated models can present instabilities when refined grid cells are present at the model boundaries. Therefore, care needs to be taken when applying dynamic downscaling. To solve this problem the first 0.3 degrees around the model domain are not being refined in this study. When changes in grid refinement are abrupt, for example due to steep bathymetry, model instabilities can also occur. The nesting of multiple models in each other would allow for a smoother grid transition and might solve this issue. Nevertheless, it is recommended not to place the model boundaries cutting topographic complex regions. Furthermore, it is to be noted that the models presented here (except G1) are uncalibrated.

Although they present an adequate performance, detailed calibration of the bed level, bottom friction and roughness
coefficients could improve the modelling results (Wang et al., 2022b).
Automated modelling tools like MOSAIC have the advantage of being efficient, reducing potential human errors and being
reproducible and transparent. However, they also have their limitations. Users must be aware of the underlying modelling
assumptions, and should carefully review the model outputs of their specific case study (Remmers et al., 2024).
### 4.3    Directions for future research
There are various directions to further develop and improve MOSAIC. In this study, we have implemented MOSAIC to
simulate coastal flooding driven by storm surges. However, flooding typically results from a combination of various drivers.
Our results underestimate flooding near estuaries and deltas due to the exclusion of precipitation and river discharge, and near
steep coasts due to the exclusion of waves and overtopping. Considering that HydroMT and SFINCS can include pluvial and
fluvial drivers (Eilander et al., 2023), there is potential to incorporate the modelling of compound events into MOSAIC. Waves
can significantly contribute to coastal flooding and, in some regions, are the dominant driver of extreme water levels (Parker
et al., 2023). However, the inclusion of wave contributions in large-scale assessments has been limited due to the computational
cost of traditional wave-resolving numerical models. The development of more computationally efficient wave solvers offers
an opportunity to implement dynamic wave simulations into large-scale assessments and into MOSAIC. For instance, Leijnse
et al. (2024) developed an efficient solver currently being integrated within SFINCS. Furthermore, this first version of
MOSAIC makes use of offline coupling for both the local-high resolution model and the SFINCS model. However, new
software developments such as the Oceanographic Multi-purpose Software Environment (OMUSeE; Pelupessy et al., 2017)
could be used to enable online coupling, as well as  to further expand MOSAIC by allowing for coupling with other models
such as hydrological or ocean models. We envisage various directions for the future application of MOSAIC beyond the
modelling of historical coastal floods presented here. By leveraging the flexibility of MOSAIC to modify input datasets, the
modelling framework can be used to study events under historical- and climate change conditions. Furthermore, taking
advantage of MOSAIC's multiscale modelling approach, TC/ETC high-resolution hazard assessments can be obtained
globally. When linked to impact models, such as Delft-FIAT (Slager et al., 2016), MOSAIC could also be used for risk
assessments.
### 4.4    Added value of the MOSAIC framework
The main added value of MOSAIC is it flexibility to simulate anywhere in the world water levels and coastal flooding with
customizable datasets and resolutions, enabling efficient, region-specific storm event simulations. Users of MOSAIC can
easily simulate storm events in any region with this modelling framework. First, they can select the appropriate meteorological
forcing. Within MOSAIC, users can choose gridded meteorological data from reanalysis datasets or climate models to simulate
ETCs or TCs, provided that the data accurately captures the TC wind and pressure fields (as seen with ETC Xynthia and TC
Irma in this study). Alternatively, they can select a hybrid approach that combines the Holland model with ERA5 in the
background when modelling smaller TCs with rapid intensification (such as TC Haiyan in this study). Depending on the
specific storm simulated and study area, users can select different model refinements. For rapidly intensifying storms, users
can choose a more refined temporal output resolution, while nested models can help resolving the topography and bathymetry
in regions with complex coastlines. If the users have coastal boundary conditions available, MOSAIC can automatically
generate stand-alone local high-resolution Delft3D FM models without having to couple them with GTSM. Although
uncalibrated, these model configurations demonstrate similar performance to the well-established global model GTSM, but at
a significantly lower computational cost. The hydrodynamic flood modelling part of MOSAIC offers user-defined settings as
well, enabling users to, for instance, choose the most suitable DEM for their study area or implement flood protection measures
through MOSAIC's HydroMT component.

## 5    Concluding remarks

The MOSAIC modelling framework introduced in this study allows to dynamically simulate coastal flooding events through the coupling of dynamic water level and overland flood models, making use of a Python environment. This approach is automated and reproducible, and combined with the underlying global datasets used, makes it globally applicable. MOSAIC's flexibility allows to easily simulate coastal flooding events globally, while also using local high-resolution models. Based on our results, we conclude that the refinement of the global modelling approach can significantly impact the simulation of coastal water levels and flood depths at local scale, although the differences in local settings make that there is no one-size-fits-all approach. We recommend higher temporal output resolution for rapidly intensifying TCs, spatial output refinement for regions with heterogeneous water levels and nested local models with high-resolution bathymetry, if available, for regions with complex topographies. The flexibility and ease of use of MOSAIC make it a valuable resource for users to further explore which are the optimal settings for their case study and region of interest.

**Appendix A: Supporting tables and figures**

**Table A1. Validation indicators that compare the maximum total water levels and observations of GESLA for the case studies Irma and Xynthia.**

| Irma | RMSE [m] | | | | Pearson correlation [-] | | | |
|---|---|---|---|---|---|---|---|---|
| Station | G1 | G2 | N1 | N2 | G1 | G2 | N1 | N2 |
| 1 | 0.41 | 0.41 | 0.39 | 0.40 | 0.92 | 0.92 | 0.92 | 0.92 |
| 2 | 0.28 | 0.27 | 0.25 | 0.25 | 0.98 | 0.98 | 0.98 | 0.98 |
| 3 | 0.33 | 0.33 | 0.32 | 0.33 | 0.79 | 0.78 | 0.81 | 0.79 |
| 4 | 0.27 | 0.26 | 0.21 | 0.24 | 0.96 | 0.96 | 0.96 | 0.94 |
| 5 | 0.35 | 0.35 | 0.33 | 0.31 | 0.93 | 0.93 | 0.93 | 0.93 |
| 6 | 0.18 | 0.18 | 0.17 | 0.21 | 0.98 | 0.98 | 0.98 | 0.94 |
| 7 | 0.17 | 0.17 | 0.14 | 0.14 | 0.97 | 0.97 | 0.95 | 0.95 |
| 8 | 0.39 | 0.39 | 0.42 | 0.45 | 0.92 | 0.92 | 0.90 | 0.88 |
| 9 | 0.16 | 0.16 | 0.18 | 0.10 | 0.93 | 0.92 | 0.90 | 0.96 |
| *Average* | *0.28* | *0.28* | *0.27* | *0.27* | *0.93* | *0.93* | *0.93* | *0.92* |
| *Standard deviation* | *0.09* | *0.09* | *0.10* | *0.11* | *0.06* | *0.06* | *0.05* | *0.05* |
| **Xynthia** | **RMSE [m]** | | | | **Pearson correlation [-]** | | | |
| Station | G1 | G2 | N1 | N2 | G1 | G2 | N1 | N2 |
| 1 | 0.12 | 0.13 | 0.13 | 0.13 | 1.00 | 1.00 | 1.00 | 1.00 |
| 2 | 0.27 | 0.29 | 0.22 | 0.26 | 0.99 | 0.99 | 0.99 | 0.99 |
| 3 | 0.21 | 0.20 | 0.47 | 0.61 | 0.99 | 0.99 | 0.95 | 0.91 |
| 4 | 0.20 | 0.21 | 0.19 | 0.34 | 1.00 | 1.00 | 1.00 | 0.98 |
| 5 | 0.18 | 0.18 | 0.24 | 0.25 | 1.00 | 1.00 | 0.99 | 0.99 |
| 6 | 0.34 | 0.31 | 0.49 | 0.92 | 0.99 | 0.99 | 0.98 | 0.90 |
| *Average* | *0.22* | *0.22* | *0.29* | *0.42* | *1.00* | *1.00* | *0.99* | *0.96* |
| *Standard deviation* | *0.08* | *0.07* | *0.15* | *0.29* | *0.01* | *0.01* | *0.02* | *0.04* |

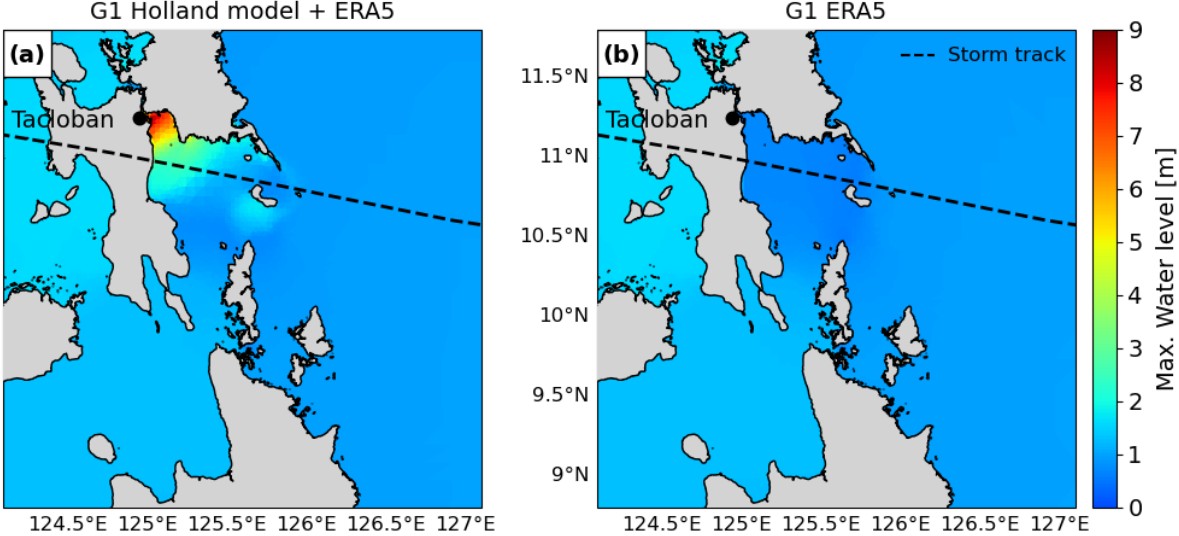


**Figure A1. Maximum water levels output of GTSM, for case study Haiyan, with different meteorological forcings. Left: Maximum**
**total water levels with the Holland model combined with ERA5 as a forcing. Right: Maximum total water levels with ERA5 as**
**forcing.**

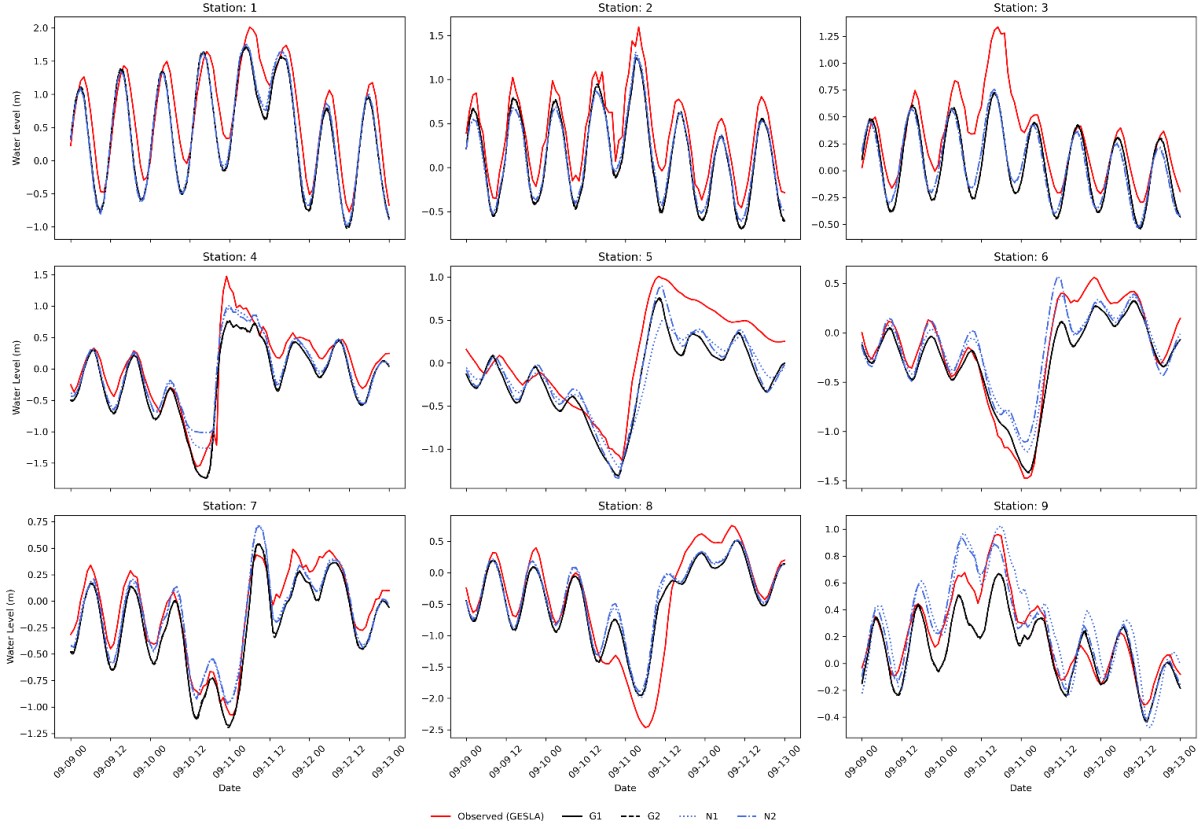


**Figure A2. Validation of total water levels for the case study Irma, for the nine locations depicted in Fig. 3.**

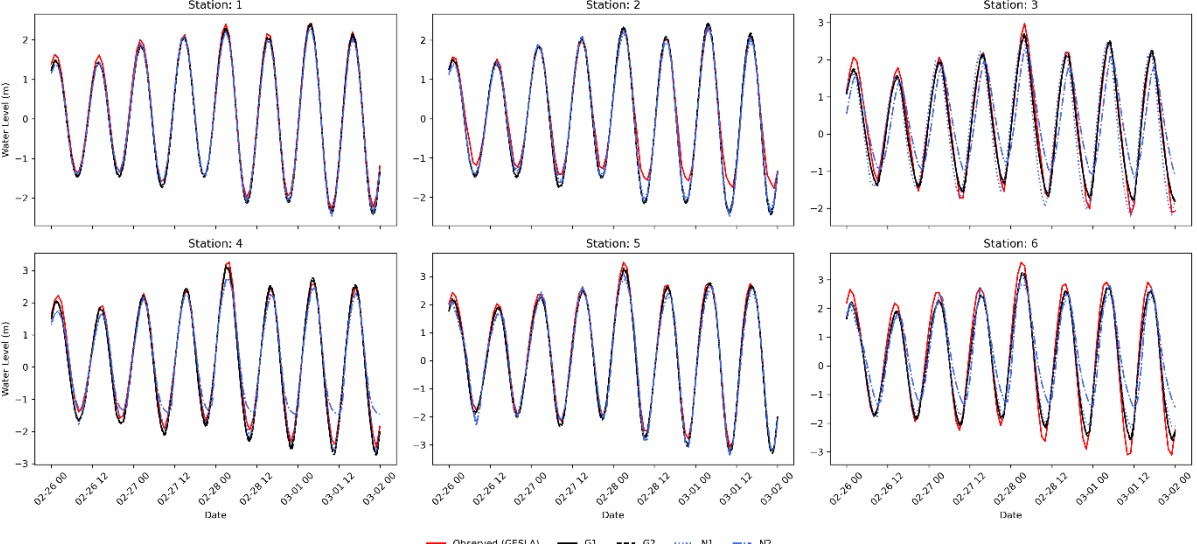


**Figure A3. Validation of total water levels for the case study Xynthia, for the six locations depicted in Fig. 3.**


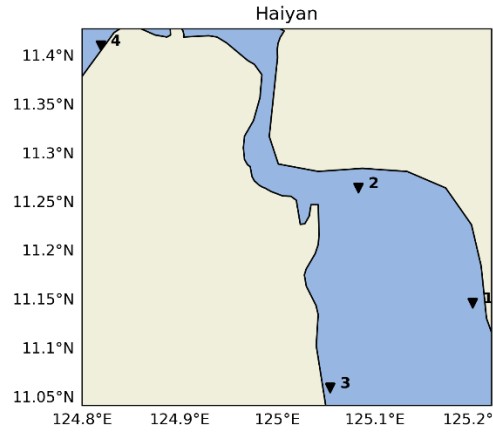


**Figure A4. GTSM output locations for the case study Haiyan.**

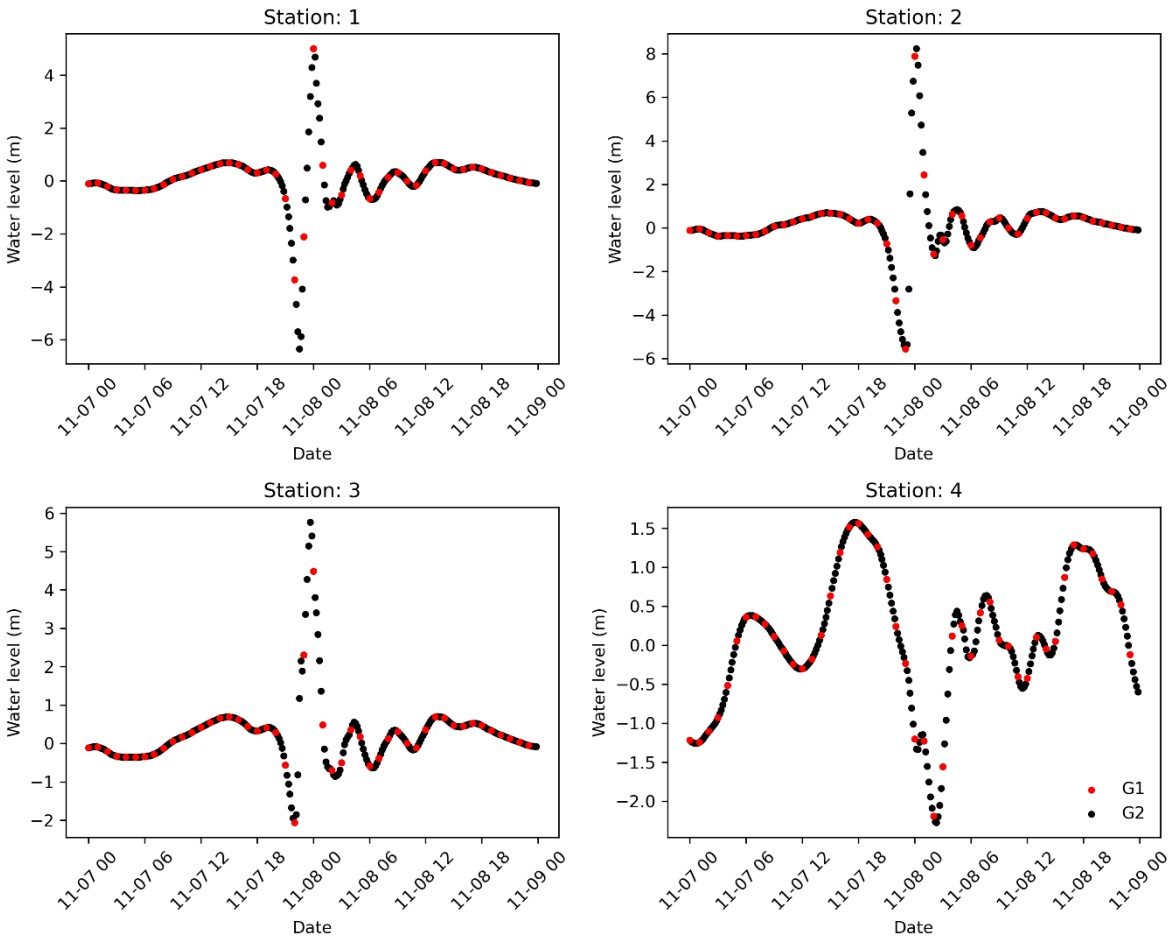


**Figure A5. Haiyan total water level timeseries for the GTSM output locations provided in Fig. A4. Timeseries for the default configuration (G1) and the refined temporal output resolution configuration (G2).**

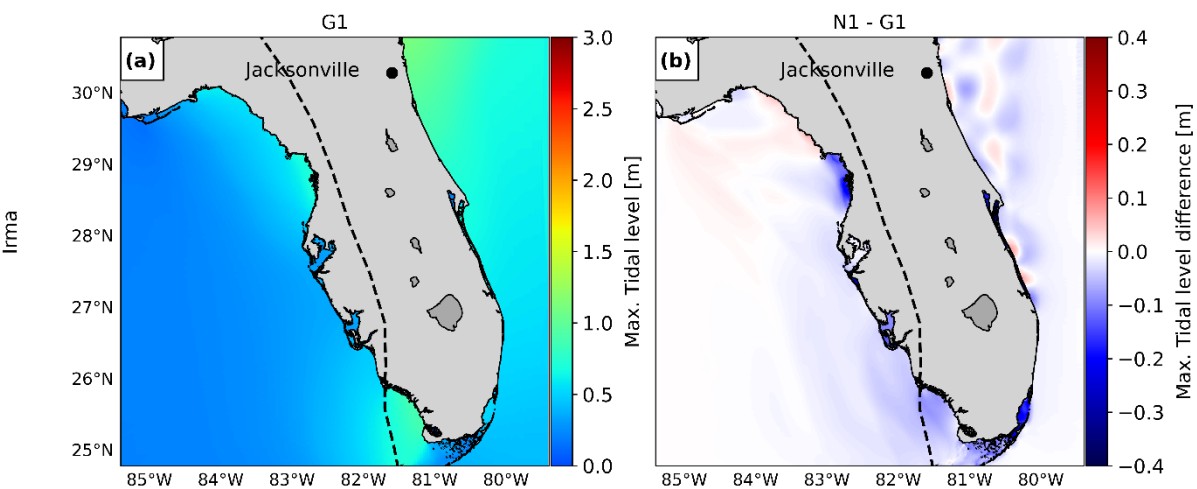


**Figure A6. Maximum water levels for the tide only simulation of G1 (panel a). Difference between the maximum water level for the tide only simulations of N1 and G1 (panel b).**

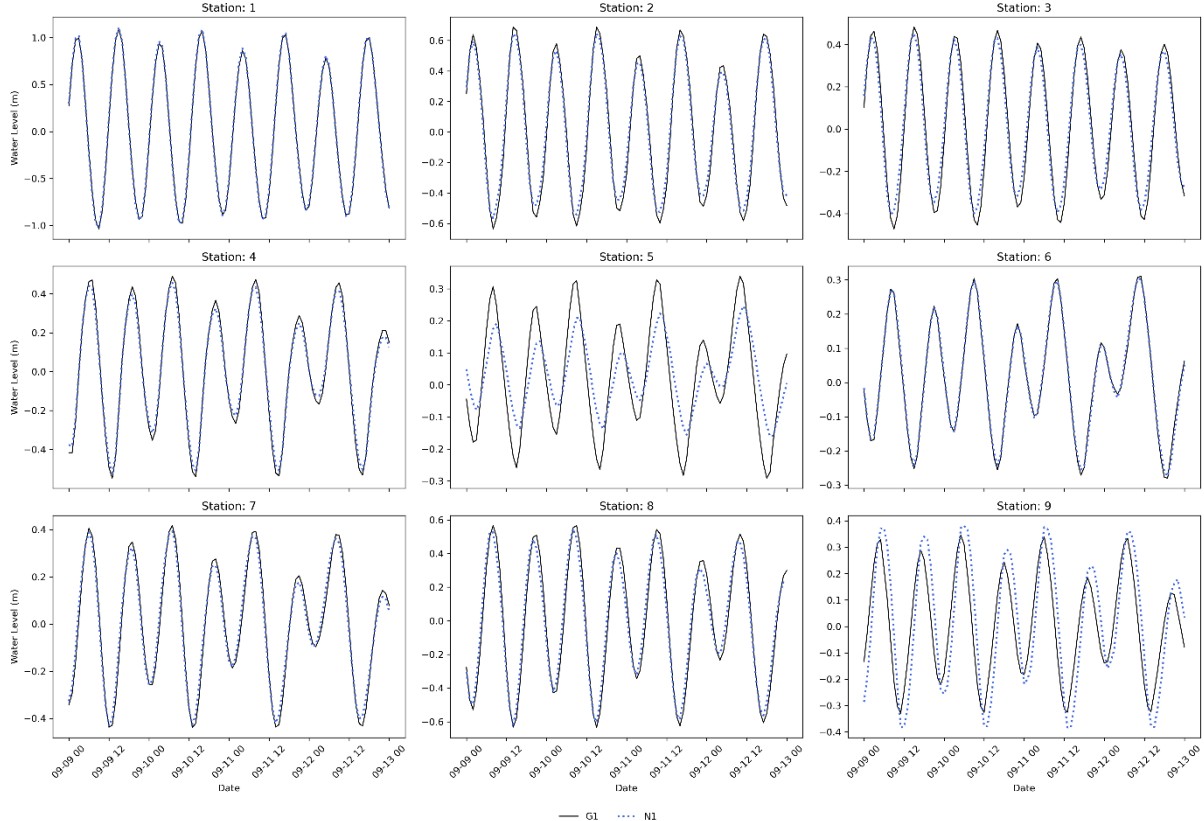


**Figure A7. Water levels for the tide only simulations for the case study Irma model configurations G1 and N1, for the nine locations depicted in Fig. 3.**



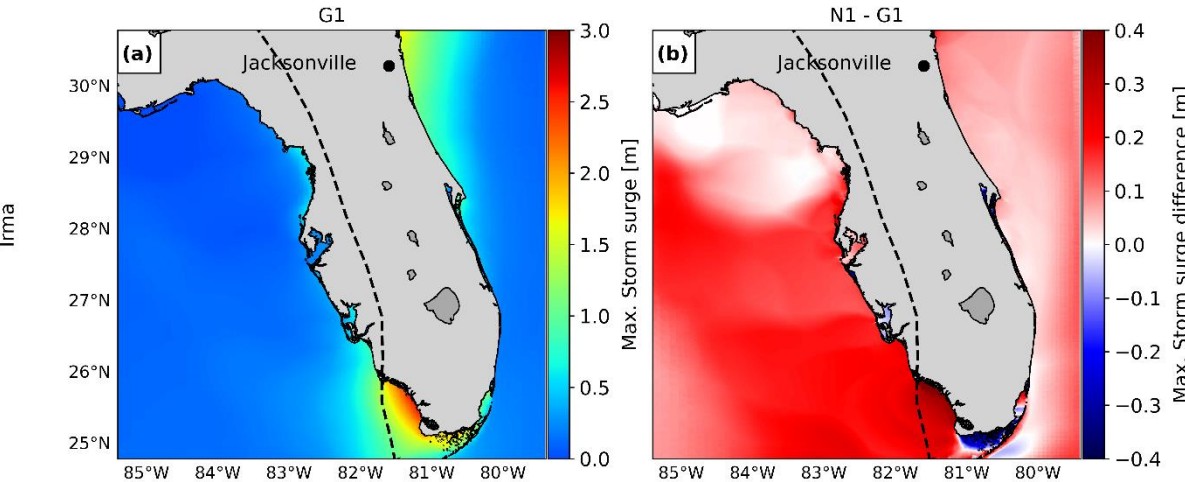


**Figure A8. Maximum water levels for the storm surge only simulation of G1 (panel a). Difference between the maximum water level**
**for the tide only simulations of N1 and G1 (panel b).**

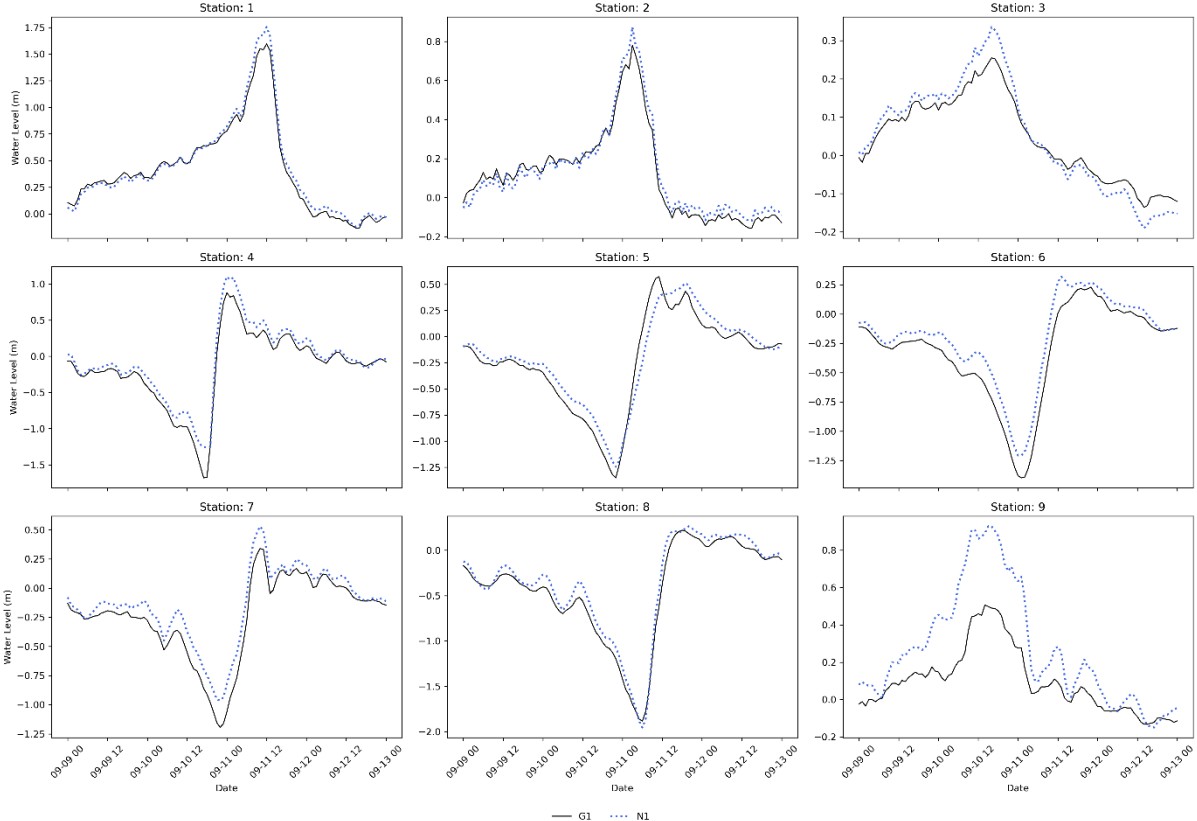


**Figure A9. Water levels for the storm surge only simulations for the case study Irma model configurations G1 and N1, for the nine**
**locations depicted in Fig. 3.**

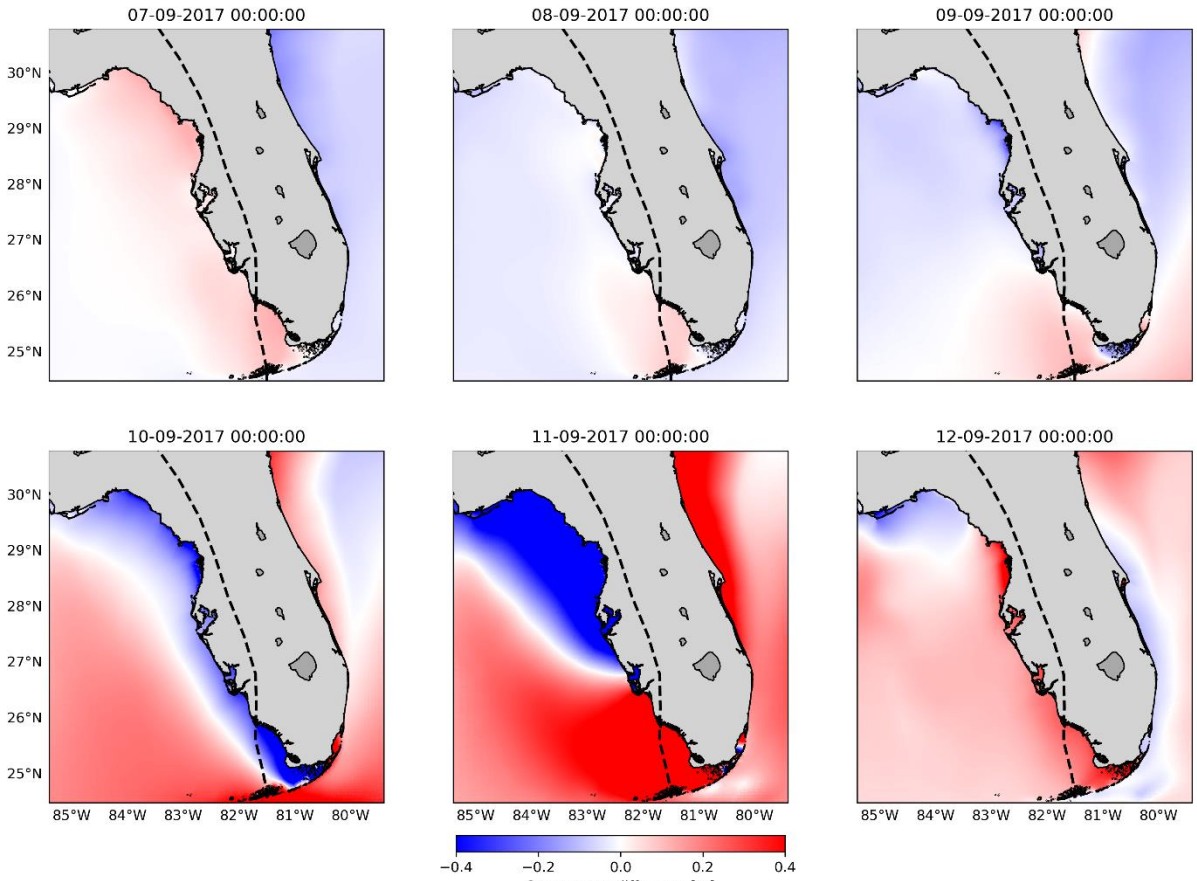

Figure A10. Difference in water levels for the storm surge only simulations of N1 and G1 for different timesteps, before TC Irma makes landfall (07-09-2017 until 09-09-2017), during the peak (between 10-09-2017 and 11-09-2017) and after the peak (12-09-2017).

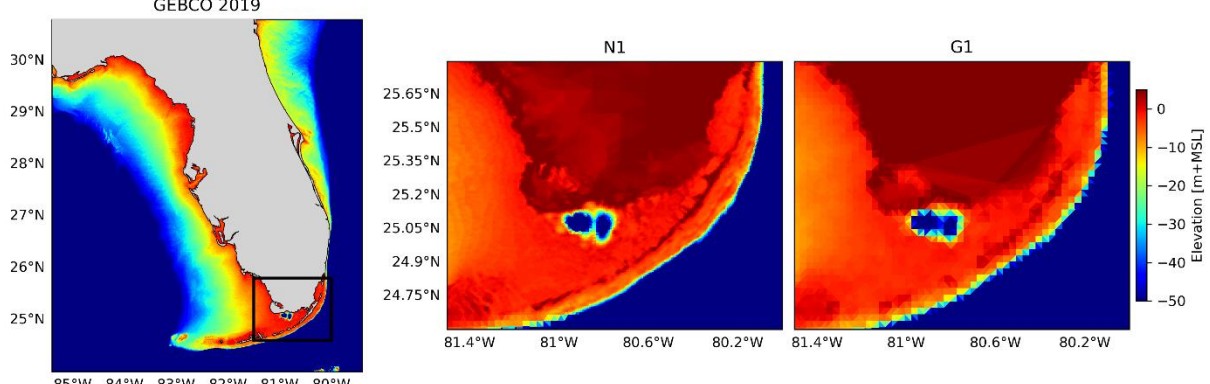

Figure A11. Left: GEBCO2019 for the study area, black rectangle shows the barrier island region from the middle and right panels. Middle: Bathymetry in the barrier island interpolated to the grid of the model configuration N1. Right: Bathymetry in the barrier island interpolated to the grid of the model configuration G1.

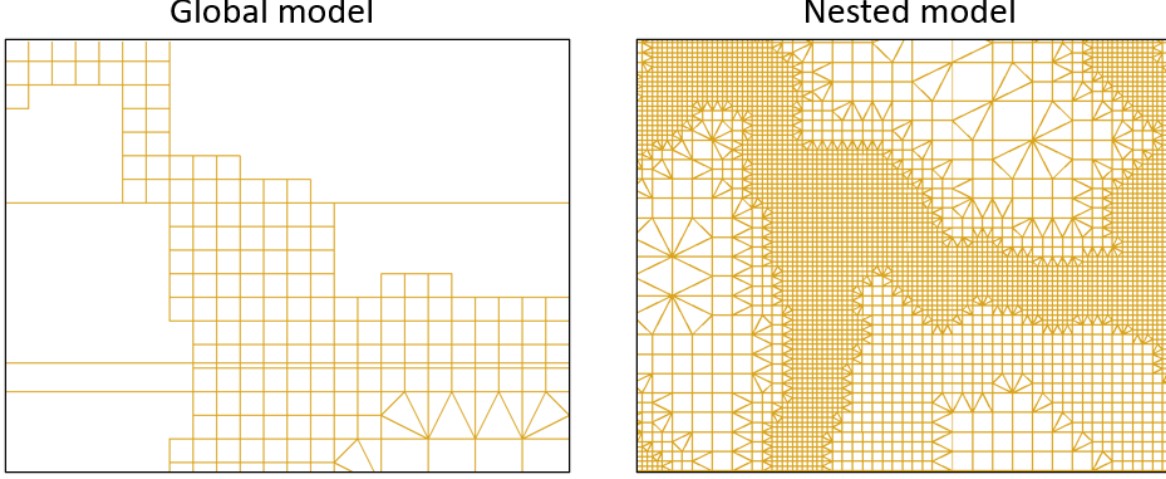

565

**Figure A12. Close look at the unstructured grid of the global GTSM model with a grid resolution up to 2.5 km along the coast (left)**

**and the nested grid of dynamic downscaling with a grid resolution up to 0.45 km along the coast (right), for case study Haiyan.**

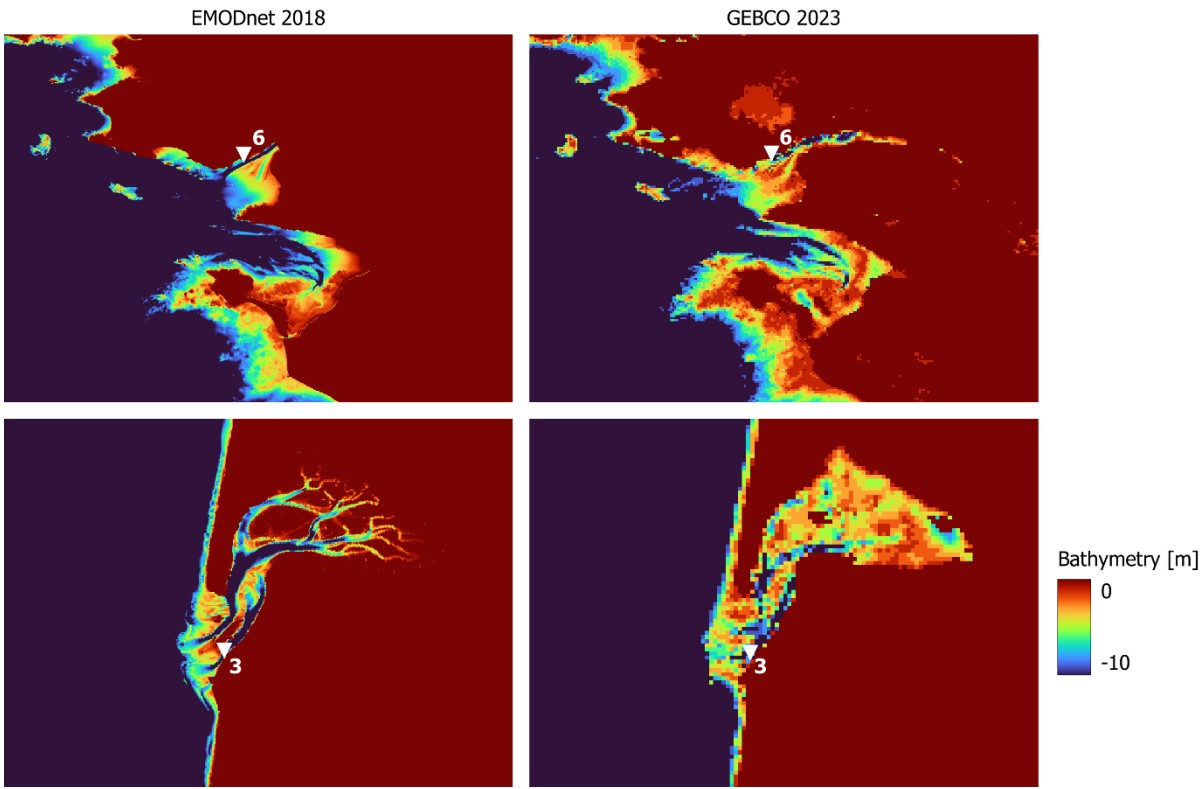

568

**Figure A13. Close look at the bathymetry of two stations (top row: station 6 and bottom row: station 3) that provide lower**

**performance with updated bathymetry, for the case study Xynthia. Left: Bathymetric map of EMODNet2018. Right: Bathymetric**

**map of GEBCO2023.**

572

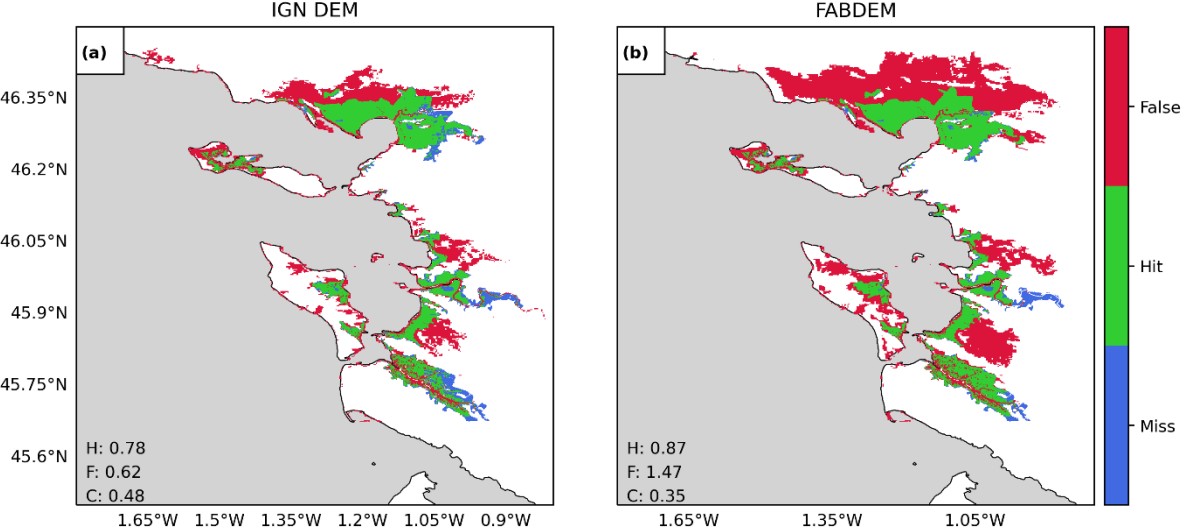

**Figure A14. Validation of flood extents for the case study Xynthia against observed flood extents. The maps compare the modelled and observed maximum flood extents for a SFINCS model generated with ING's DEM (panel a) and FABDEM (panel b), where: green indicates flood areas correctly simulated; blue flood areas not simulated but observed; and red flood areas simulated but not predicted. Performance indicators for the hit rate (H), false-alarm ratio (F) and critical success index (C) are shown in each panel.**

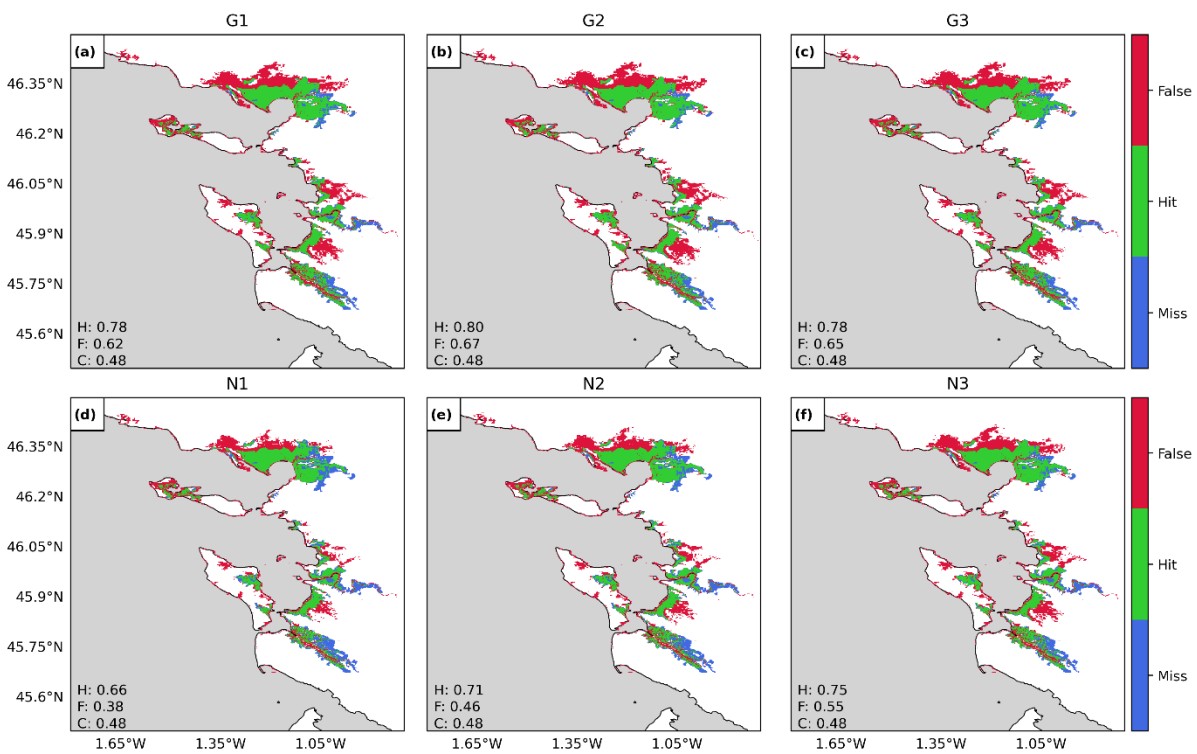

**Figure A15. Validation of flood extents for the case study Xynthia against observed flood extents. The maps compare the modelled and observed maximum flood extents for each model configuration, see Table 1, where: green indicates flood areas correctly simulated; blue flood areas not simulated but observed; and red flood areas simulated but not predicted. Performance indicators for the hit rate (H), false-alarm ratio (F) and critical success index (C) for each configuration are shown in each panel.**

## Data availability

The datasets compiled and/or analysed during the current study are available on Zenodo. *Note: to be published with Doi upon acceptance of the paper.*

## Code availability

The underlying code for this study is available on at https://github.com/Ireneben73/mosaic_framework (last access: 11 October 2024).

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

**Acknowledgements**

This work was carried out in the EU-ERC COASTMOVE project nr 884442 and the NWO MOSAIC project nr
ASDI.2018.036. The authors would like to thank the SURF Cooperative for the support in using the Dutch national e-
infrastructure under grant no. EINF-2224 and EINF-5779.

**Author contributions**

I.B.: Conceptualisation, Investigation, Methodology, Modelling, Visualisation, Analysis, Writing – Original Draft. J.C.J.H.A.:
Conceptualisation, Investigation, Methodology, Writing – Review & Editing, Supervision. P.J.W.: Conceptualisation,
Investigation, Methodology, Writing – Review & Editing, Supervision. D.E.: Conceptualisation, Investigation, Methodology,
Modelling, Writing – Review & Editing, Supervision. S.M.: Conceptualisation, Investigation, Methodology, Modelling,
Writing – Review & Editing, Supervision.
**Competing interests**
One of the (co-)authors is a member of the editorial board of Natural Hazards and Earth System Sciences.