# Peer review of "A multiscale modelling framework of coastal flooding events for global to local flood hazard assessments"

_EGUsphere, 2024_

## Author Comment (AC1)

**A multiscale modelling framework of coastal flooding events for global to local flood hazard assessments**

**Responses to reviewers:**

**Reviewer #1:**

Dear Authors,

After reviewing your manuscript, I find that it holds significant relevance and potential. However, in its current form, it fails to address any substantial scientific question. The results presented are merely a model-model comparison, illustrating that variations in spatial or temporal resolution affect the outcomes. This does not address the scientific questions or limitations introduced in the introduction. Therefore, I recommend major revisions and provide the following comments to help strengthen the manuscript and enhance its scientific rigor.

*We are pleased to read that the reviewer finds our paper has potential to provide a valuable scientific contribution. In response to the detailed and constructive comments, we have significantly revised the paper.*

*We have rewritten the introduction to clearly outline the limitations addressed by the manuscript. In the manuscript we now present a modelling framework that is easy to use and flexible, and we have conducted a sensitivity analysis to examine the impacts of model resolution on the simulation of total water levels and flood extents. We have also included validation of the total water levels and flood extents into the manuscript to strengthen the foundation of our modelling framework.*

*We believe this has greatly improved the manuscript and we thank the reviewer for the time taken to review our manuscript.*

1. **Introduction**: This section is well-written but could benefit from more specifics regarding the dynamic processes that are currently missing (L46-51). For instance, details on wave-driven processes, hydrological processes, and man-made structures would be valuable. Additionally, I challenge the notion that the limitation of topo-bathymetry in global applications can be resolved solely through grid refinement. In my mind, there are three main methodological challenges: resolution, input data sets (topo-bathy and others), and physical processes. This paper addresses the first one but not the other two. Hence, the linkage from the scientific gap to the approach does not hold, as the MOSAIC modeling framework does not resolve challenges with input data nor does it address additional processes relevant for inundation that global models fail to account for.

*Thank you for the insightful suggestions regarding the introduction. Following your recommendation, we have restructured the main methodological challenges into three sections. Additionally, we have added the inclusion of multiple flood drivers as a challenge for flood risk assessments. Finally, the*

*manuscript has been revised to explicitly state that our primary aim is to benchmark the implications of model resolution:*

[revised manuscript text omitted]

2. **MOSAIC Modeling Framework**: The authors aim to introduce a modeling framework. To do so successfully, a more comprehensive introduction to other modeling frameworks and/or nesting techniques is necessary. Additionally, more details are needed on what has been specifically programmed and what is novel about it. For example, details on the Holland parametric wind model and how it is integrated are missing. I also miss details on the nesting procedure used for both the offline Delft3D FM and SFINCS approach. A more rigorous description of the code would enhance the scientific value of the manuscript.

*Thank you for your valuable suggestions. In response to your comments, we have included more information about current modelling frameworks. Specifically, we now present a modelling framework that uses model nesting to improve the outputs in ocean models, and extends to land to hydrodynamically simulate compound flooding. The manuscript has been updated accordingly as follows:*

*(Lines 52 – 54): "Grid refinement and nesting of local high-resolution models within coarser global models can result in improved coastal boundary conditions. Pelupessy et al. (2017) used a similar multiscale approach to obtain realistic boundary conditions by nesting a global circulation model and a high-resolution barotropic model."*

*(Lines 62 – 70): "Additionally, large-scale hazard assessments typically focus on a single flood driver (Tiggeloven et al., 2020; Vousdoukas et al., 2018b; Ward et al., 2020). However, TC and ETC events often produce precipitation, river discharge, storm surges and waves, all of which can contribute to flooding. When these drivers occur in combinations, they can significantly amplify flood hazards and risks. For instance, recent research showed that storm surge exacerbates fluvial flooding at global scale (Eilander et al., 2020). Few studies have analysed the effects and interactions of multiple flood drivers. While Bates et al. (2021) performed a combined risk assessment of fluvial, pluvial and coastal flooding for the continental USA, Eilander et al. (2023) introduced the first globally-applicable compound flood modelling framework that accounts for precipitation, river discharge and storm tides. However, the inclusion of waves in large-scale assessments and the interactions between flood drivers remains a challenge."*

*Using the Holland model is an option within the modelling framework that can be used when the user considers it necessary. To enhance the clarity on the Holland parametric wind model, we have added the following lines to the manuscript, where we define more details of the Holland model and how it is used in the framework:*

*(Lines 103 – 113): "Because TC Haiyan is not well resolved in ERA5 (see Fig. A1), we use pressure and wind from tropical cyclone track data merged with ERA5. The tropical cyclone track data is retrieved from the Joint Typhoon Warning Center at 6 hourly intervals (Naval Meteorology and Oceanography Command, 2022) and is converted to a polar grid with 36 radial bins, 375 arcs and a radius of 350 km using the Holland parametric wind model (Holland et al., 2010). Following the methodology of Dullaart et al. (2021) and Lin and Chavas (2012), we apply a counter-clockwise rotation angle of β = 20° and set the storm translation to surface background wind reduction factor at α = 0.55. Additionally, we use an empirical surface wind reduction factor (SWRF) of 0.85 (Batts et al., 1980), and convert 1-minute average winds to 10-minute averages using a factor of 0.915 (Harper et al., 2010). The Holland model's output provides a file that defines a polar grid containing pressure and wind fields. To extend the pressure and wind fields beyond the Holland model's defined TC boundary, we linearly interpolate these fields on the outermost 75% to align with the ERA5 background data (Deltares, 2024)."*

*In order to improve the explanation of the nesting procedure used for nesting GTSM and the local Delft3D FM model, we have updated the manuscript as follows:*

*(Lines 152 – 157): "Second, MOSAIC uses an offline coupling approach to nest the local Delft3D Flexible Mesh model within GTSM. A Python script is used to first identify the boundaries of the local Delft3D Flexible Mesh model. These boundaries are then used to determine the specific locations where GTSM output should be extracted. Subsequently, GTSM provides the water level timeseries at the boundaries of the local model. Finally, the local high-resolution model is executed using the water levels derived from GTSM as forcing input, together with the same meteorological forcing as for GTSM."*

*Although we previously mentioned using HydroMT to couple GTSM and SFINCS, the specific output from GTSM that serves as input for SFINCS was not clearly explained. To address this, we have updated the manuscript with the following clarifications:*

*(Lines 125 – 126): "GTSM provides as output water level timeseries over a grid in the ocean and for locations along every ~5 km of the coast."*

*And:*

*(Lines 175 – 182): "To build the SFINCS models and couple them with GTSM, MOSAIC uses the HydroMTv0.7.1 (Hydro Model Tools) package (Eilander et al., 2023). HydroMT is an open-source Python package, which provides automated and reproducible model building and analysis of results. HydroMT uses a modular approach in which datasets and model setup configurations can easily be interchanged. In the MOSAIC framework presented in this paper, we take advantage of HydroMT in several ways: (1) to automatically convert the forcing files from GTSM and the other input into the model specific input format; (2) to easily build a reproducible SFINCS model; and (3) to perform the analysis of the SFINCS model output. SFINCS is forced with GTSM water level timeseries at locations along every ~5 km of the coastline, and provides as output water level timeseries for each grid cell. Finally, flood depth maps are derived from the maximum water levels by subtracting the DEM elevation."*

3. **Modeling Results**: This section requires the most work. As mentioned, in its current state, it is a model-model comparison without any significant insights. To address this, I strongly recommend the authors include relevant observations of observed water levels and flood extents. Without these, it is difficult to assess differences and model accuracy.

*Thank you for the suggestions. We understand the concern about including relevant observations of water levels and flood extents. For this reason, we have thoroughly revised this section and included a validation of the results of the case studies for which we had observations. We also improved the sensitivity analysis and provided additional results of this analysis in the appendix. Overall, we believe the result section is much stronger now. Note that the model validation does not provide a conclusive result in terms of which model configuration provides the best results. This is really case dependent, and it is difficult to validate a global model with the limited tide gauge stations available. Nevertheless, no model configuration shows bad results, and the flexibility and easiness of MOSAIC can be used by users to explore which settings are best for their specific case study and region of interest. We elaborate on the changes made in the manuscript below.*

*We have validated the total water levels using observations from the GESLA tide gauge stations for the case studies Irma and Xynthia (Haigh et al., 2023). Thanks to this more thorough validation, we decided to update the GTSM model version from 3 to 4.1, which has a better tidal performance. Furthermore, we also decided to use ERA5 only for Irma, as this showed better results than the Holland model, which overestimated the peak of the event. For TC Haiyan, on the other hand, ERA5 alone did not capture the TC and therefore, we used the Holland model combined with ERA5 at the background. We updated the manuscript as follows:*

[revised manuscript text omitted]

*This shows that for ETC Xynthia the G@ models can correctly represent the flooded areas due to higher hit rates, but overestimate the flooding more in the northern regions, while the N@ models underestimate the flooding in the south due to the lower water levels simulated in that region.*

[Figure]

**Figure A9. Validation of flood extents for the case study Xynthia against observed flood extents. The maps compare the modelled and observed maximum flood extents for each model configuration, see Table 1, where: green indicates flood areas correctly simulated; blue flood areas not simulated but observed; and red flood areas simulated but not predicted. Performance indicators for the hit rate (H), false-alarm ratio (F) and critical success index (C) for each configuration are shown in each panel.**

The insights regarding the relevance of temporal and spatial resolution require more simulations to assess convergence. For example, the authors could show differences between 1, 2, 5, 10, 20, and 60 minutes of temporal resolution to demonstrate how water levels respond to these changes. A similar approach can be taken for spatial resolution. During these comparisons, please avoid varying the bathymetry source simultaneously, as this would complicate the findings. When performing these analyses, provide an analysis that supports the findings. Why are water levels higher or lower with these settings?

*Thank you for this good suggestion. While it would be highly valuable to explore the convergence of our findings across different temporal resolutions, the computational demands of such an analysis is very high and prevent us from performing this test. However, as the reviewer suggested, we have introduced an additional model configuration that isolates the effects of grid refinement without simultaneously altering the bathymetric dataset. These updates are now reflected in the manuscript as follows:*

*(Lines 206 – 220): "Using the MOSAIC modelling framework, we analyse the effects of refining the resolution of GTSM on the simulated water levels and assess how these propagate into the results for the flood hazard simulated by SFINCS. As described in Table 1, we categorise model configurations in two distinct groups. The first group, which contains the global model configurations (G), includes the default model configuration (G1) and configurations that modify only the global GTSM model (G2 and G3). In this group, the refinements applied are: (1) the temporal output resolution, which is different than the implicitly calculated simulation*

*timestep of GTSM, is refined from 1-hourly to 10-minute, allowing to capture more changes in water levels, including the peaks of the water levels (G2); and (2) the spatial output resolution is refined from locations along the coast every ~5 km to ~2 km, providing more coastal boundary conditions for the hydrodynamic flood hazard model (G3). The second group, which contains the nested model configurations (N), includes those model configurations that use a nested local model within the global model GTSM by performing dynamic downscaling. These model configurations include: (1) the nesting of local high-resolution models with refined grids into GTSM (N1); and (2) the nesting of local high-resolution models with refined grids and updated bathymetry into GTSM (N2). Finally, we evaluate the combined effects of all these refinements through the "fully refined" configuration (N3), which integrates both the enhanced temporal and spatial resolutions as well as the nested high-resolution models and updated bathymetry. The validation of GTSM and SFINCS shows sufficient performance for all the model configurations from Table 1 and Fig. 7 (see Table A1 and Figs. A2 and A3)."*

*Table 1. GTSM model configurations used in the sensitivity analysis.*

| Model configuration | Nomenclature | GTSM grid resolution | Bathymetry | Spatial output resolution | Temporal output resolution |
|---|---|---|---|---|---|
| *Default configuration* | *G1* | *~25 to 2.5/1.25km* | *GEBCO2019\** | *Original (~5 km)* | *1h* |
| *Refined temporal output resolution* | *G2* | *~25 to 2.5/1.25km* | *GEBCO2019\** | *Original (~5 km)* | *10min* |
| *Refined spatial output* | *G3* | *~25 to 2.5/1.25km* | *GEBCO2019\** | *Refined (~2 km)* | *1h* |
| *Dynamic downscaling (Refined grid)* | *N1* | *~25 to 0.45km* | *GEBCO2019\** | *Original (~5 km)* | *1h\*\** |
| *Dynamic downscaling (Refined grid + Updated bathymetry)* | *N2* | *~25 to 0.45km* | *GEBCO2023* | *Original (~5 km)* | *1h\*\** |
| *Fully refined configuration* | *N3* | *~25 to 0.45km* | *GEBCO2023* | *Refined (~2 km)* | *10min\*\** |

*\* EMODnet2018 for Europe (Xynthia case study)*

*\*\*For the model configurations N1, N2 and N3, the temporal output resolution is also the temporal resolution of the coupling between GTSM and the local high-resolution model.*

*We have also added a dedicated section in the results to analyze these effects. Subsequently, we have examined the impact of updating to a new bathymetry within the refined grid. In the results we have also interpreted why changes in the model configuration result in higher or lower water levels. These updates are now reflected in the manuscript as follows:*

*(Lines 258 – 294):*

[revised manuscript text omitted]

I am particularly skeptical about the dynamic downscaling/fully refined results. How do the authors explain a 40 cm increase in water level? It seems there might be a double-counting of the inverse barometer effect or another error. I do not believe that the entire Gulf of Mexico can have such a different water level based on minor model configuration changes. Could the authors provide more justification for these findings? To understand the results better, I recommend analyzing the time series first.

*When validating the total water levels of Irma, Figure A2 presented above shows that specially for the southwest of Florida, while the default configuration G1 presents a slight underestimation of the peaks overall, the dynamic downscaling (N1 and N2) might overestimate those peaks slightly. Therefore, the ground truth is somewhere in between both modelling results, and it does not mean that G1 underestimated the peaks by 40 cm. When looking at table A1 presented above, both model configurations actually show similar results, with G1 having a RMSE of 0.28 m and a Pearson's correlation of 0.93, and N1 and N2 a RMSE of 0.27 and a Pearson's correlation of 0.93 and 0.92 respectively.*

In this section, the authors use the word 'might' frequently. I suggest analyzing the results to test these hypotheses. For example, why are the results different for Haiyan with a 60-minute temporal resolution? One can demonstrate this by comparing water levels near the eye of the storm and further away, providing results rather than hypotheses.

In response to this and other comments from both reviewers, the result section was rewritten and significantly changed. We believe that these textual changes and additional analyses address the concerns of the reviewer. Specifically for Haiyan, we have included in the manuscript more results that help on the interpretation of the results:

*(Lines 242 – 247): "For TC Haiyan (Fig. 8 panel f), the sensitivity of the water levels is significant. Water levels increase due to the temporal refinement up to 2 m along the coastlines where TC Haiyan made landfall, showing that 1-hourly resolution is too coarse to accurately capture the water level response. The cause for this is that TC Haiyan had a rapid intensification, and when modelling water levels at 1-hourly resolution we overlook the storm's peak, resulting in an underestimation of the maximum water levels. G2 however, can capture the peak of TC Haiyan more precisely (see Figs. A4 and A5)."*

[Figure]

**Figure A4. GTSM output locations for the case study Haiyan.**

[Figure]

**Figure A5. Haiyan total water level timeseries for the GTSM output locations provided in Fig. A3. Timeseries for the default configuration (G1) and the refined temporal output resolution configuration (G2).**

In the flood section, the results are unconvincing. For example, during Irma, Jacksonville experienced severe flooding. In Figure 7 (a-d), the city appears unaffected. This is problematic. I suspect that topo-bathymetry is the cause, which brings us back to the challenges mentioned in the introduction that MOSAIC does not resolve. Demonstrating 1) accurate water levels near the city and 2) flood extents with more reliable US-based topo-bathymetry are essential to successfully model this case study.

Indeed the topography can play a key role in modelling flood depths and extents. Moreover, the flooding in Jacksonville was largely due to heavy precipitation. In this first version of our MOSAIC modelling framework we do not include multiple flood drivers and focus only on the surge as driver of coastal flooding.  As a result of the validation suggested by the reviewer, we decided to update the DEM used for the case study Xynthia. In this region there were many dikes that prevented the water from travelling further inland. However, the DEM used before, FABDEM, could not resolve those. With the updated DEM from IGN, we can resolve better the dikes and obtain more accurate results. Nevertheless, the best approach for this would be to integrate, when possible, the flood protection measures in the hydrodynamic flood model. We have updated the manuscript as follows to integrate this:

*(Lines 166 - 169): "For this study, we use GEBCO 2020 (15 arc seconds spatial resolution; (Weatherall et al., 2020)) as input dataset for the bathymetry and FABDEM (30 m spatial resolution; (Hawker et al., 2022)) as input dataset for the land elevation. Except for ETC Xynthia. For ETC Xynthia we use the 5 m resolution LiDAR-based DEM developed by the French National Geographic Institute (IGN) because it better represents dikes in the region, leading to better flood estimates than FABDEM (see Fig. A8)."*

[Figure]

***Figure A8. Validation of flood extents for the case study Xynthia against observed flood extents. The maps compare the modelled and observed maximum flood extents for a SFINCS model generated with ING's DEM (panel a) and FABDEM (panel b), where: green indicates flood areas correctly simulated; blue flood areas not simulated but observed; and red flood areas simulated but not predicted. Performance indicators for the hit rate (H), false-alarm ratio (F) and critical success index (C) are shown in each panel.***

4. **Discussion**: I could not find the MOSAIC code on Zenodo, so I argue that this needs to be shared first before claiming it is 'automated and reproducible.' I also challenge the statement "enhance the simulation at the local scale by providing refined water levels." I have not seen evidence of this in the manuscript.

We have added the github link to MOSAIC in the manuscript. The datasets compiled during the study will be available on Zenodo upon acceptance of the paper:

*(Lines 488 – 490): "**Code availability***

*The underlying code for this study is available on at https://github.com/Ireneben73/mosaic_framework*

*(last access: 11 October 2024)."*

---

## Author Comment (AC2)

**A multiscale modelling framework of coastal flooding events for global to local flood hazard assessments**

**Responses to reviewers:**

**Reviewer #2:**

This manuscript presents a methodology for evaluating flooding at high resolution by coupling three models: GTSM, Delft3D, and SFINCs. These models are among the latest and most robust developments in hydrodynamic modeling. On one hand, it is necessary to develop robust methodologies to assess coastal flooding, taking into account different types of forcings such as tropical cyclones (TCs) and extratropical cyclones (ETCs). On the other hand, this study tries to emphasize the importance of increasing both temporal and spatial resolution as well as enhancing hydrodynamic flood modeling.

Therefore, this research and development could be a valuable contribution to the scientific community. However, I believe this study fails to convincingly demonstrate that refining, downscaling, and dynamic flood modeling significantly improve flood hazard assessment. Several aspects need clarification, and a clear message about your results, conclusions, or recommendations for performing a flood risk assessment has not been adequately addressed.

*We are pleased that the reviewer recognises the potential scientific contribution of our manuscript. In response to the detailed and constructive feedback, we have made substantial revisions to the manuscript.*

*We have rewritten the introduction to clearly outline the limitations addressed by our study, presenting a modelling framework that is easy to use and flexible. To do so, we have conducted a sensitivity analysis to examine the impacts of model resolution on the simulation of total water levels and flood extents. We have also included validation of the total water levels and flood extents into the manuscript to strengthen the foundation of our modelling framework. It is important to note that from the model validation we cannot conclude that refinements always lead to an improved model performance. Instead, each refinement is case specific and therefore the user of the modelling framework should decide depending on the type of storm and location of the analysis which refinement is most adequate.*

*We believe that these amendments have greatly enhanced the manuscript and now we better answer the question how downscaling affects the model performance. We thank the reviewer for the time and effort dedicated to reviewing our work.*

The weakest part of your study is the sensitivity analysis of the model configurations.

First, concerning the organization, it would be advisable to assign a nomenclature to each configuration to aid in comparisons and analysis. For example, the default configuration and the refined temporal and spatial output could be assigned the same letter with different numbering (since they all originate from the same model, with the same forcing and bathymetry). The fully refined configuration should have another letter, because although it combines higher temporal and spatial resolution, the fact that the GTSM simulations use more current and detailed bathymetry (despite Europe EDMOnet being used

with the same resolution as GEBCO2014) introduces another distinct element that can significantly affect the results. Finally, the dynamic downscaling (a nomenclature related to the global configuration, to which the fully refined configuration is nested), should not be directly compared with the default configuration without analyzing the effect of the previous factors.

This distinction between configurations should also be used for comparing results at the storm surge modeling and hydrodynamic modeling levels. This way, the analyses would be more orderly, identifying how each factor influences the outcomes, leading to more conclusive comparisons and differences.

In section 3.1 (multiscale storm surge modeling), I would divide it into three subsections: the first analyzing the effect of higher resolution on the maximum water level value (results shown in Figure 5 b, e, and h), the second identifying the effect of bathymetry changes (moving from GEBCO2014 to GEBCO2023, results shown in Figure 4), once the time resolution effect is identified, and the third extracting the added value of dynamic downscaling (once the effects of time resolution and bathymetry changes are understood).

*Thank you for your valuable suggestions. In response to these suggestions, we have provided a nomenclature for each model configurations, distinguishing between model configurations that are or modify the global GTSM model (G), and model configurations that include a nesting of a locally refined model (N). Furthermore, for a more stepwise sensitivity analysis of the dynamic downscaling, we have separated the process into two model configurations: first, grid refinement alone retaining the old GEBCO2019 bathymetry (N1), followed by a dynamic downscaling with updated bathymetry interpolated to GEBCO2023 (N2). We chose this approach, rather than starting with the updated bathymetry alone and then applying grid refinement, because the latter could be automated in Python, while the former required manual intervention. This stepwise approach allows to isolate the effects of the grid refinement from those of updating the bathymetry. Furthermore, as suggested by the reviewer we have split the section into subsections that identify the effects of higher resolution, changed in grid refinement and bathymetry.*

*We have updated the manuscript as follows:*

*(Lines 206 – 220): "Using the MOSAIC modelling framework, we analyse the effects of refining the resolution of GTSM on the simulated water levels and assess how these propagate into the results for the flood hazard simulated by SFINCS. As described in Table 1, we categorise model configurations in two distinct groups. The first group, which contains the global model configurations (G), includes the default model configuration (G1) and configurations that modify only the global GTSM model (G2 and G3). In this group, the refinements applied are: (1) the temporal output resolution, which is different than the implicitly calculated simulation timestep of GTSM, is refined from 1-hourly to 10-minute, allowing to capture more changes in water levels, including the peaks of the water levels (G2); and (2) the spatial output resolution is refined from locations along the coast every ~5 km to ~2 km, providing more coastal boundary conditions for the hydrodynamic flood hazard model (G3). The second group, which contains the nested model configurations (N), includes those model configurations that use a nested local model within the global model GTSM by performing dynamic downscaling. These model configurations include: (1) the nesting of local high-resolution models with refined grids into GTSM (N1); and (2) the nesting of local high-resolution models with refined grids and updated*

*bathymetry into GTSM (N2). Finally, we evaluate the combined effects of all these refinements through the "fully refined" configuration (N3), which integrates both the enhanced temporal and spatial resolutions as well as the nested high-resolution models and updated bathymetry. The validation of GTSM and SFINCS shows sufficient performance for all the model configurations from Table 1 and Fig. 7 (see Table A1 and Figs. A2 and A3)."*

*We have also added the nomenclature in the table below:*

**Table 1. GTSM model configurations used in the sensitivity analysis.**

| Model configuration | Nomenclature | GTSM grid resolution | Bathymetry | Spatial output resolution | Temporal output resolution |
|---|---|---|---|---|---|
| *Default configuration* | *G1* | *~25 to 2.5/1.25km* | *GEBCO2019\** | *Original (~5 km)* | *1h* |
| *Refined temporal output resolution* | *G2* | *~25 to 2.5/1.25km* | *GEBCO2019\** | *Original (~5 km)* | *10min* |
| *Refined spatial output* | *G3* | *~25 to 2.5/1.25km* | *GEBCO2019\** | *Refined (~2 km)* | *1h* |
| *Dynamic downscaling (Refined grid)* | *N1* | *~25 to 0.45km* | *GEBCO2019\** | *Original (~5 km)* | *1h\*\** |
| *Dynamic downscaling (Refined grid + Updated bathymetry)* | *N2* | *~25 to 0.45km* | *GEBCO2023* | *Original (~5 km)* | *1h\*\** |
| *Fully refined configuration* | *N3* | *~25 to 0.45km* | *GEBCO2023* | *Refined (~2 km)* | *10min\*\** |

*\* EMODnet2018 for Europe*

*\*\*For the model configurations N1, N2 and N3, the temporal output resolution is also the temporal resolution of the coupling between GTSM and the local high-resolution model.*

*We have added a dedicated section in the results to analyse these effects. Subsequently, we have examined the impact of updating to a new bathymetry within the refined grid. In the results we have also interpreted why changes in the model configuration result in higher or lower water levels. These updates are now reflected in the manuscript as follows:*

*(Lines 258 – 294):*

[revised manuscript text omitted]

In section 3.2 (Hydrodynamic flood modeling), I would also make a stepwise comparison, isolating the effect of different factors (spatial resolution, temporal resolution, bathymetry, and dynamical downscaling). In the first subsection, compare the default configuration with the refined temporal resolution and refined spatial scale. In the second subsection, compare the results of the fully refined configuration with the dynamical downscaling to isolate and identify this effect. The goal is to see what each factor contributes and how high-resolution modeling with SFINCs improves, once the effect of water level (as the boundary condition) is accounted for. Comparing the default configuration with the dynamical downscaling does not reveal which factor is more influential, for example, if a more detailed bathymetry in the global model is already the most determining factor.

*We have done this separation in the revised manuscript and included a more stepwise analysis of the results for the dynamic downscaling. We have explained this in the previous comments, together with the changes of section 3.1.*

It would be beneficial to conclude with some recommendations, as this methodology can be applied by various researchers and consultants in their flooding studies. It should be noted that the available

water level data are related to the default configuration. Therefore, it is crucial to identify the added value of each factor rather than jumping directly to the dynamical downscaling nested in the refined spatial output. Other researchers applying this methodology would start from the water level of the default configuration, even without modeling the TC forced by the Holland wind model (as in the Global Sea Level time series available at Copernicus Climate Data Store: https://doi.org/10.24381/cds.a6d42d60). How well or how it may affect using the Holland model?

*Thank you for the suggestion. Adding more information and suggestions on how the modelling framework could be used by researchers does help to better understand the applications that MOSAIC could be used for. We have modified the manuscript as follows:*

*(Lines 466 – 483): "Users of MOSAIC can easily simulate storm events in any region with this the modelling framework. First, they can select the appropriate meteorological forcing. Within MOSAIC, users can choose gridded meteorological data from reanalysis datasets or climate models to simulate ETCs or TCs, provided that the data accurately captures the TC wind and pressure fields (as seen with ETC Xynthia and TC Irma in this study). Alternatively, they can select a hybrid approach that combines the Holland model with ERA5 in the background when modelling smaller TCs with rapid intensification (such as TC Haiyan in this study). Depending on the specific storm simulated and study area, users can select different model refinements. For instance, the G2 model configuration with refined temporal output resolution is suitable for rapidly intensifying storms, while nested models can resolve help resolving the topography and bathymetry in regions with complex coastlines. If the users have coastal boundary conditions available, MOSAIC can automatically generate stand-alone local high-resolution Delft3D FM models (N1 , N2, and N3 model configurations) without having to couple them with GTSM. Although uncalibrated, these model configurations demonstrate similar performance than the well-established global model GTSM (G1; see Section 3), but at a significantly lower computational cost. The hydrodynamic flood modelling part of MOSAIC offers user-defined settings as well, enabling users to, for instance, choose the most suitable DEM for their study area or implement flood protection measures through MOSAIC's HydroMT component.*

*By leveraging the flexibility of MOSAIC to modify input datasets, the modelling framework can be used to study events under historical- and climate change conditions. Furthermore, taking advantage of MOSAIC's multiscale modelling approach, TC/ETC high-resolution hazard assessments can be obtained globally. When linked to impact models, MOSAIC can also be used for risk assessments."*

Another significant drawback of your study is the lack of validation with observations of water levels and flood extents. Is there any possibility to validate the steps of the MOSAIC methodology with observations of water levels or spatial flooding maps? How can we be sure that increasing spatial/temporal resolution, improving bathymetry, and employing dynamic downscaling enhance the results?

*Thank you for the valuable suggestions and valid questions. We understand the concern about including relevant observations of water levels and flood extents. For this reason, we have validated the modelling framework results for the case studies for which observations are available.*

*We have validated the total water levels using observations from the GESLA tide gauge stations for the case studies Irma and Xynthia. Thanks to this more thorough validation, we decided to update the GTSM model version from 3 to 4.1, which has a better tidal performance. Furthermore, we decided to use ERA5 only for Irma, as this showed better results than the Holland model, which overestimated the peak of the event. For TC Haiyan, on the other hand, ERA5 alone did not capture the TC and therefore we used the Holland model combined with ERA5 in the background. We updated the manuscript as follows:*

[revised manuscript text omitted]

The effects of waves (especially infragravity energy), precipitation, and river discharge should be addressed and discussed more thoroughly in the introduction (other methodologies that take into account these factors) and in the discussion (what effects could be missed) .

*We have included a reflection on the flood drivers that are considered in this manuscript and the ones that could be applied later on within MOSAIC:*

*(Lines 62 – 70): "Additionally, large-scale hazard assessments typically focus on a single flood driver (Tiggeloven et al., 2020; Vousdoukas et al., 2018b; Ward et al., 2020). However, TC and ETC events often produce precipitation, river discharge, storm surges and waves, all of which can contribute to flooding. When these drivers occur in combinations, they can significantly amplify flood hazards and risks. For instance, recent research showed that storm surge exacerbates fluvial flooding at global scale (Eilander et al., 2020). Few studies have analysed the effects and interactions of multiple flood drivers. While Bates et al. (2021) performed a combined risk assessment of fluvial, pluvial and coastal flooding for the continental USA, Eilander et al. (2023) introduced the first globally-applicable compound flood modelling framework that accounts for*

*precipitation, river discharge and storm tides. However, the inclusion of waves in large-scale assessments and the interactions between flood drivers remains a challenge."*

*(Lines 456 – 462): "In this study, we have implemented MOSAIC to simulate coastal flooding driven by storm surges. However, since flooding typically results from a combination of various drivers, our results currently underestimate flooding near estuaries and deltas due to the exclusion of precipitation and river discharge, and near steep coasts due to the exclusion of waves and overtopping. Future research on TCs and ETCs may further develop MOSAIC and include other drivers such as waves, rainfall and discharge. Considering that HydroMT and SFINCS are capable of handling compound flooding induced by pluvial and fluvial drivers (Eilander et al., 2023), there is potential for future enhancements of MOSAIC to incorporate the modelling of compound events."*

Specific question:

Local high resolution model domain: why these domains have been selected? Have they been defined based on the cyclone tracks?

*Yes, these domains were selected based on the tropical cyclone tracks, and also considering that the size of the model domain was not too small, given that that could cause model instabilities.*

Minor comments:

GEBCO 2020 in line 119 while GEBCO2023 is mentioned in Table 1.

*GEBCO2020 is used for the flood hazard modelling using SFINCS, while the updated bathymetry of GTSM is GEBCO2023 (this is what Table 1 refers to).*

Realistic configuration in Figure 8: not sure what result/output is this.

*This was a typo. We have updated the figure with the new nomenclatures:*

[Figure]

**Figure 12. Flood depth timeseries for three observation points and flood volume timeseries for the SFINCS model domain of each case study. The spatial location of the SFINCS output point locations can be observed in Fig. 11 panels a, d, g.**

---

## Author Response (AR2)

**A multiscale modelling framework of coastal flooding events for global to local flood hazard assessments**

**Responses to reviewers:**

**Reviewer #1:**

Dear Authors,

I want to start by congratulating you on the significant improvements to your manuscript. It's clear that a great deal of time and effort has gone into addressing the previous round of feedback, and I commend your commitment to refining your work. Below, I provide my detailed comments following the structure from my earlier review:

*Dear reviewer, thank you for your kind words and acknowledging the effort put into improving the manuscript. Your comments were really helpful in getting a better version of it.*

1. Introduction

The revisions to the introduction largely address my earlier concerns, and I appreciate the improvements. However, I still feel that waves should be more explicitly described since they are integral to many coastal studies. Especially their absence in the discussion leaves a critical gap in the context of the study.

*Dear reviewer, thank you for your words and suggestions. We have revised the manuscript to better highlight the importance of waves and how those could be addressed in future studies.*

> *(Lines 541 – 546): "Waves can significantly contribute to coastal flooding and, in some regions, are the dominant driver of extreme water levels (Parker et al., 2023). However, the inclusion of wave contributions in large-scale assessments has been limited due to the computational cost of traditional wave-resolving numerical models. The development of more computationally efficient wave solvers offers an opportunity to implement dynamic wave simulations into large-scale assessments. For instance, Leijnse et al. (2024) developed an efficient solver currently integrated within SFINCS, which could potentially be implemented into future iterations of the MOSAIC modelling framework."*

2. MOSAIC Modeling Framework

While additional details about the framework have been included, the manuscript still feels incomplete.
• For a more comprehensive introduction, I recommend referencing other established modeling frameworks and techniques in the introduction. For instance, the CoSMoS framework by Barnard et al. (2014) should be mentioned, as it was among the first to address similar challenges.

*Thank you for the suggestion. To provide additional context on modelling frameworks that address similar challenges to those discussed in this manuscript, we have included the CoSMoS framework in the introduction:*

*(Lines 54 - 56): "Similarly, Barnard et al. (2014) developed a framework that nests dynamically downscaled global tide and wave models with local cross-shore profile and cliff failure models."*

• Additionally, more specifics are needed about how the nesting is performed. For example: I presume that the authors are using water levels for boundary conditions, correct? And what boundary condition types are applied in the local models SFINCS? In general, I think it would be good to address the limitations of water level nesting. For instance, the Riemann boundary could be a more appropriate choice in certain cases. In general, a detailed explanation of these technical aspects is essential to enhance the manuscript's clarity and rigor.

Thank you for your comments. In the manuscript, we explain the boundary conditions used to nest the Delft3D FM model within GTSM. The local Delft3D FM models use water level outputs from the global model GTSM as boundary conditions:

*(Lines 154 - 159): "Second, MOSAIC uses an offline coupling approach to nest the local Delft3D Flexible Mesh model within GTSM. A Python script is used to first identify the boundaries of the local Delft3D Flexible Mesh model. These boundaries are then used to determine the specific locations where GTSM output should be extracted. Subsequently, GTSM provides the water level timeseries at the boundaries of the local model. Finally, the local high-resolution model is executed using the water levels derived from GTSM as forcing input, together with the same meteorological forcing as for GTSM."*

The boundary conditions for SFINCS are water level timeseries from GTSM, as described in the manuscript as follows:

*(Lines 182 – 184): "SFINCS is forced with GTSM water level timeseries at locations along every ~5 km of the coastline, and provides as output water level timeseries for each grid cell. Finally, flood depth maps are derived from the maximum water levels by subtracting the DEM."*

Riemann boundary conditions could be more suitable in certain cases. In such cases, Riemann boundary conditions can be selected as an option within Delft3D FM.

3. Modeling Results

The added validation is a welcome improvement, providing a stronger foundation for the findings. I particularly appreciate (and follow) the description of the temporal component (G1 to G2).

However, I remain surprised by the significant water level changes (exceeding 10 cm) in the Gulf of Mexico due to changes in input bathymetry (G1 to N1). This requires further explanation. Could the tidal propagation with the updated GEBCO bathymetry be driving these changes or is there something else? For example, a separate analysis of tide and surge components could help identify the source. – In the case of N1, the GEBCO bathymetry used is GEBCO2019, which is the same as used for G1, with the only difference being the grid refinement. The differences in water levels in this region seem to come from N1's higher resolution, which better captures the complex topographic features in southwest Florida, such as the barrier islands. Additionally, the finer grid resolution in N1 results in a more detailed representation of the bathymetry, even though both models use the same underlying

bathymetric data. This enhanced resolution allows N1 to better capture significant changes in bed level, particularly in shallow regions of southwest Florida. These local bathymetric differences may help explain the variation in storm surge during TC Irma, as storm surges are significantly sensitive to bathymetry, particularly in shallow regions. For instance, at Stations 4 and 9 in Figure A2, prior to Irma's landfall, water levels are similar across all model configurations. This suggests that the differences in the simulations do not come from the tide simulations. On the other hand, once TC Irma makes landfall the differences in water levels, caused by the storm surge, start deviating for the two model configurations G1 and N1.

Regarding Hurricane Irma, I question the suitability of calling the results "good" with an RMSE of 40 cm. Given the availability of higher-accuracy bathymetry and topography sources for this US case study, it is surprising these were not utilized. This could be a missed opportunity to enhance model performance, as you demonstrate in the European case study. Is there are reason I am missing why those data sources are not utilized?

Improved bathymetric data could certainly enhance the performance of the storm surge model. However, as the primary objective of this manuscript was to present a flexible modelling framework for multiscale modelling that allows to select from different datasets, rather than to create a perfect simulation, we chose not to update the bathymetry. Nevertheless, MOSAIC is designed to easily update bathymetry, allowing for future updates in subsequent research.

4. Discussion

Thank you for sharing the code used in the study.

5. References

I noticed several instances of what appear to be lazy or potentially incorrect referencing. While self-publication is a natural part of academic work, I feel it is crucial to engage broadly with the wider scientific community to ensure a balanced and representative citation list.

For example:

• Bloemendaal et al. (2019) and Dullaart et al. (2020) are cited as evidence that "errors in bathymetric datasets propagate to storm surge modeling." From my understanding, these papers do not directly make this claim. A more suitable reference might be Parodi et al. (2020). – Thank you for the suggestion. We have modified the references and moved the reference to Dullaart to the effects in spatial and temporal resolution while adding Parodi et al. in the references to the bathymetry effects in water levels.

*(Lines 56 – 58): "Coarse meteorological forcings – both in terms of spatial and temporal resolution – might not be able to capture the resolution necessary to resolve intense storms (Dullaart et al., 2020), while errors in the bathymetric datasets will propagate to the modelling of storm surge levels(Parodi et al., 2020)"*

• Similarly, Eilander et al. (2023) are credited with introducing the first globally applicable compound

flood modeling framework that accounts for precipitation, river discharge, and storm tides. However, van Ormondt et al. (2020) appear to have done something similar earlier. – I believe the reviewer is referring to the paper introducing the Delft Dashboard. While Delft Dashboard has proven to be a really useful and interactive tool for retrieving inputs for various hydrodynamic models and even simulate overland flooding, the flooding modelled in this manuscript is solely driven by water levels induced by a tsunami, rather than by multiple compound flood drivers. For this reason it has not bee included as a reference to frameworks that perform compound flood modelling.

I would recommend the authors to carefully review all their citations to ensure that they are appropriate.

Thank you for the suggestion. We have reviewed the references of the manuscript to make sure they are appropriate.

Minor Comments

• The additional technical details on tropical cyclones are appreciated. However, please include, the wind drag coefficients and the background pressure used in this study. – Thank you for the suggestions. We have added those parameters into the manuscript:

*(Lines 122 – 125): "Tides are generated internally with tide generating forces, while storm surges originate from external forcing with pressure and wind fields. A constant Charnock coefficient of 0.041 is applied to translate wind speeds from the external forcing into wind drag, and a background pressure of 101,325 Pa is considered."*

• Avoid using the term "total water level" unless wave contributions are explicitly accounted for. A more accurate term for this study is "still water level." – Thank you for the suggestion. We have modified "total water level" to "water level", where in lines 116-117 of the manuscript we define water levels as the result from tides and storm surges: "MOSAIC uses GTSMv4.1 to simulate water levels resulting from tides and storm surges, ignoring baroclinic and wave contributions."

• A typographical error in Figure 1: "Delft3d" should be "Delft3D" (with a capital D). - Corrected

I hope these comments are helpful in further improving your manuscript. Thank you for the opportunity to review this important work, and I look forward to seeing its next iteration.

---

## Author Response (AR3)

Dear Authors,

Thank you for submitting the revised manuscript. I appreciate your detailed point-by-point responses. However, I must note that several of my comments from the previous round have not been adequately addressed. To facilitate further improvements, I will focus on the most critical aspects that require attention.

Dear reviewer, thank you for spending the time to review the manuscript and providing feedback. We have considered your suggestions and used them to further improve the manuscript. The main changes include the following:

- Better explanation of what drives the water levels differences between the model configurations G1 and N1;
- Better interpretation of the meaning of the RMSE values and comparison against other large-scale studies;
- Restructuring of the discussion sections and rewriting some other parts of the manuscript to better highlight the added value of our work;
- Updating of references to better reflect the latest state-of-the-art modelling approach for large-scale storm surge and flood assessments.

Below we provide response to your comments and include the changes that have been made in the manuscript.

3. Modeling Results:

I acknowledge the progress made in addressing some aspects of the modeling results, particularly the temporal component (G1 to G2), which is clear and logical. However, I remain very concerned about the significant water level changes (over 10 cm) observed in the Gulf of Mexico due to changes in input bathymetry (G1 to N1). Neither the revised manuscript nor the rebuttal provides a sufficient justification for these findings. Phrases such as "may help explain" are speculative and do not constitute a robust explanation.

Dear reviewer, thank you for your acknowledgment of the improvement in the temporal component results. Regarding the differences between G1 and N1, we have conducted further analyses based on your feedback, to better understand the source of these differences. As you suggested, to investigate the mechanism behind the water level changes we have examined the tidal and storm surge components separately.

*Tidal contribution*

To analyse the tidal contribution, we have ran tide-only simulations for both model configurations without meteorological forcing. Figure A6 shows the differences in maximum tide-only water levels between G1 and N1, indicating no significant differences occur in the Gulf of Mexico, with variations only along the coast. Figure A7 further confirms this at the observation stations, where most locations exhibit minimal

differences between the two configurations. The only exceptions are stations 5 and 9. At station 5, the difference comes from its location in an estuary, while at station 9, the difference is caused by the interpolation of bathymetry from N1 onto a higher-resolution grid, allowing better representation of the barrier islands and affecting the tidal range (see Figure A11).

[Figure]

*Figure A6. Maximum water levels for the tide only simulation of G1 (panel a). Difference between the maximum water level for the tide only simulations of N1 and G1 (panel b).*

[Figure]

*Figure A7. Water levels for the tide only simulations for the case study Irma model configurations G1 and N1, for the nine locations depicted in Fig. 3.*

*Surge contribution*

To analyse the storm surge contribution, we have ran surge-only simulations with meteorological forcing but without tidal forcing. Figure A8 shows that differences between G1 and N1 occur primarily due to the storm surge propagations. The timeseries of surge-only simulations (Figure A9) and spatial differences between G1 and N1 per timestep (Figure A10) show that the differences between G1 and N1 do not persist throughout the event, but are most pronounced at the peak of TC Irma.

[Figure]

*Figure A8. Maximum water levels for the storm surge only simulation of G1 (panel a). Difference between the maximum water level for the tide only simulations of N1 and G1 (panel b).*

[Figure]

*Figure A9. Water levels for the storm surge only simulations for the case study Irma model configurations G1 and N1, for the nine locations depicted in Fig. 3.*

[Figure]

*Figure A10. Difference in water levels for the storm surge only simulations of N1 and G1 for different timesteps, before TC Irma makes landfall (07-09-2017 until 09-09-2017), during the peak (between 10-09-2017 and 11-09-2017) and after the peak (12-09-2017).*

Two main factors influence storm surge propagation: (1) meteorological forcing and (2) bathymetry.

(1) Regarding the meteorological forcing, we have looked at the differences due to the interpolation of ERA5 data into the grid of G1 and N1. Figures R1, R2 and R3, show that pressure and wind fields remain almost identical between the two configurations, indicating that atmospheric forcing is not responsible for the observed discrepancies.

[Figure]

*Figure R1. MSL pressure for the storm surge only simulations for the case study Irma model configurations G1 and N1, for the nine locations depicted in Fig. 3.*

[Figure]

*Figure R2. X component of the wind vector for the storm surge only simulations for the case study Irma model configurations G1 and N1, for the nine locations depicted in Fig. 3.*

[Figure]

*Figure R3. Y component of the wind vector for the storm surge only simulations for the case study Irma model configurations G1 and N1, for the nine locations depicted in Fig. 3.*

(2) When analysing the bathymetry, significant differences arise from the interpolation of GEBCO 2019 onto the grids of G1 and N1. The sharp changes in bathymetry along the coasts of Florida (see Figure A11, left panel), lead to interpolation differences that can exceed 100 m in depth at the same location. Despite both configurations using the same source of bathymetry, GEBCO 2019, differences in resolution result in substantial different depth representations.

In the barrier islands south of Florida, for instance, the coarser G1 grid does not fully resolve the barrier island topography (Figure A11, left panel), while the finer N1 grid captures these features more accurately (Figure A11, middle panel). These differences in bathymetry have a direct impact on storm surge propagation. Figure A10 illustrates that in the finer-resolution N1 configuration, water can pass through the barrier islands more effectively than in G1. At timestep 10-09-2017, when there is a negative surge north of the barrier island, G1 produces higher water levels because the water remains trapped. Conversely, during the peak of TC Irma on the 11-09-2017, the water levels in G1 are lower than N1 because less water can travel northward. The increased northward surge in N1 propagates further into the Gulf of Mexico, leading to the observed differences between the two configurations.

This effect of the barrier island can also be observed in Figure 4. At station 9, located on the barrier island, N1 produces a higher storm surge peak than G1, resulting in simulated peak water levels closer to the observations. Similarly, at station 4, situated north of the barrier island, N1's peak water levels resemble better the observed peak than G1.

[Figure]

*Figure A11. Left: GEBCO2019 for the study area, black rectangle shows the barrier island region from the middle and right panels. Middle: Bathymetry in the barrier island interpolated to the grid of the model configuration N1. Right: Bathymetry in the barrier island interpolated to the grid of the model configuration G1.*

[Figure]

*Figure 4. Validation of water levels for the case study Irma, for the nine tide gauge stations depicted in Fig. 3.*

We have updated the manuscript to clarify the key aspects discussed above:

(Lines 271 – 287): *"For TC Irma (Fig. 8 panel c), the nesting of a local model at high-resolution with GEBCO2019 results in maximum water levels that are up to 0.3 m higher than G1 in the southwest of Florida. These differences between N1 and G1 gradually increase over time and are maximum at the peak of TC Irma (Fig. A10). While higher grid resolution affects tidal propagation mainly along the coast of Florida (Fig. A6 and Figure A7), storm surge propagation is more sensitive to the used bathymetry (Fig. A8 and Figure A9). High resolution is needed in areas with steep bathymetry. In contrast to the coarser grid of G1, N1 better resolves complex topographic features around the barrier islands (Fig. A11), allowing water to flow more freely through these barriers. At timestep 10-09-2017 in Figure A10, when there is a negative surge north of the barrier island, G1 produces higher water levels because water remains trapped in the north. Conversely, during the peak of TC Irma, on the 11-09-2017, the water levels in G1 are lower than N1 because less water is able to travel northwards. The increased northward surge of N1 propagates further into the Gulf of Mexico, leading to higher water levels that also propagate further into the Gulf of Mexico (see Figure A10). Water levels for nine tide gauge stations along the coast indicate that while G1 underestimates the peak of TC Irma in most locations (Fig. A2, all stations but station 7), N1 simulates on average higher peaks, resulting sometimes in overestimations (Fig. A2, station 9). The improved resolution of topographic features in the barrier island region allows stations nearby (Fig. A2, stations 4 and 9) to better capture the event's peak compared to G1. Additionally, the performance of N1 is slightly better than G1 for six tide gauge stations (stations 1-6), as reflected in Table A1, which shows lower RMSE values. However, for stations 7-9, G1 shows slightly higher RMSE and Pearson's correlation."*

Furthermore, the characterization of the Hurricane Irma results as "good" is questionable given an RMSE of 40 cm. I strongly recommend that you address these points with additional analysis and provide a clearer, evidence-based justification in the manuscript.

We acknowledge the reviewer's concern regarding the RMSE of 40 cm for one station of Hurricane Irma. However, we would like to clarify the overall model performance. The Pearson's correlation for TC Irma across all model configurations is around 0.9, demonstrating that the model captures the hydrographs of the event well. Additionally, the average RMSE across all stations is 0.28 m for model configuration G1, which aligns with previous large-scale studies that have reported RMSE values of storm events ranging from 0.24 m to 0.31 m (Gori et al., 2023; Marsooli & Lin, 2018; Vogt et al., 2024). It is important to note that these studies also present RMSE averaged over all stations, rather than individual stations.

The specific case of the Fernandina Beach station (station 1 in the manuscript), where the RMSE reaches 0.41 m, is consistent with the results from Leijnse et al. (2021), who simulated water levels using a nested local model for Jacksonville within a regional model of Florida. While their study did not explicitly report RMSE values, visual comparisons suggest similar performance of their SFINCS and Delft 3D models, as illustrated in the figure below.

A key factor influencing the model's performance is its ability to predict the peak water levels of the storm. As observed in Figure 4, the model tends to underestimate water levels around the peak of the event. This is primarily due to the use of ERA5 forcing data, which do not resolve tropical cyclones with the same level of detail as higher-resolution meteorological datasets. Additionally, the model does not account for wave contributions, which play a significant role in southern Florida. For instance, at Station 3, significant wave heights during TC Irma have been modeled to exceed 10 m (Xian et al., 2018), and the absence of wave contributions in our water levels can explain those differences. Furthermore, the location of certain stations affects the validations results. Many of the stations with higher RMSE values are situated behind barrier islands or semi-enclosed bays, areas where GTSM is known to have limitations in resolving water levels (Muis et al., 2019).

Overall, while some stations show higher RMSE values, our results are within the range of similar large-scale studies. Given the high correlation of approximately 0.9, the model successfully captures the event's overall dynamics, and differences in the peaks could be further improved by users of MOSAIC when updating the meteorological forcing or including wave dynamics. However, validation of stations located in places with difficult topography will remain challenging.

[Figure]

*Figure from Leijnse et al., 2021. The hydrograph of Fernandina Beach modelled by Leijnse et al., 2021 with Delft3D and SFINCS shows very similar performance to that we present in Figure 4 above (station 1).*

We have updated the manuscript to clarify the validation of our results:

(Lines 143 – 150): "*Additionally, TC Irma has an average RMSE of 0.28 m with a standard deviation of 0.09 m. ETC Xynthia has a RMSE of 0.22 m with a standard deviation of 0.08 m. The stations performing less well are those located in enclosed harbours or behind the barrier islands. The RMSE values of GTSM for both storms show results comparable to other large-scale studies that have used hydrodynamic models to simulate storm tides of storm events. Marsooli and Lin (2018) and Gori et al. (2023), for example, used the ADvanced CIRCulation model (ADCIRC) to simulate storm tides with an average RMSE over stations of 0.31 and 0.29 m, respectively. Vogt et al. (2024) used the GeoCLaw solver and reported an average RMSE of 0.24 m over 213 tide gauge stations, but with a Pearson's correlation of 0.5, showing less good agreement with observed storm tides than the MOSAIC model setup presented in this study.*"

5. References:

I have noticed a recurring issue with references that needs to be addressed. While I appreciate that some of my suggested references were incorporated, the broader review of citations appears incomplete. In several cases, the citations remain incorrect or overly reliant on self-publication. For example, the description of van Ormondt et al. (2020) as solely addressing a tsunami is inaccurate. As noted in the abstract, the paper discusses three case studies: tides in the North Sea, storm surge and wave modeling under tropical cyclone conditions, and tsunami simulation. This misrepresentation suggests insufficient engagement with the cited literature. Such oversights undermine the scientific rigor of the manuscript. A thorough review of all references is essential to ensure accurate representation and a meaningful engagement with the broader scientific community. Without these improvements, it is difficult to support the manuscript in its current form.

Thank you for your feedback. We appreciate your observation regarding the citations and we have carefully reviewed and updated them. On the other hand, flood modelling is a very active research field with numerous regional studies and it is impossible (nor the aim of this paper) to provide a complete overview, so we have attempted to include the most important key papers of the coastal modelling community that are relevant to our work. In particular, we have incorporated additional findings and references from other research groups throughout the introduction, methods, results and discussion sections to ensure a more comprehensive and accurate representation of the current state of the art.

Below we include a list of updated references:

[revised manuscript text omitted]

I hope this feedback helps guide your revisions. I look forward to seeing the manuscript further refined and strengthened.